# Causal Spatio-Temporal Prediction: An Effective and Efficient Multi-Modal Approach

**Yuting Huang, Ziquan Fang,**\* **Zhihao Zeng, Lu Chen, Yunjun Gao**
Zhejiang University
{huangyuting, zqfang, zengzhihao, luchen, gaoyj}@zju.edu.cn

## Abstract

Spatio-temporal prediction plays a crucial role in intelligent transportation, weather forecasting, and urban planning. While integrating multi-modal data has shown potential for enhancing prediction accuracy, key challenges persist: (i) inadequate fusion of multi-modal information, (ii) confounding factors that obscure causal relations, and (iii) high computational complexity of prediction models. To address these challenges, we propose $E^2$-CSTP, an **E**ffective and **E**fficient **C**ausal multi-modal **S**patio-**T**emporal **P**rediction framework. $E^2$-CSTP leverages cross-modal attention and gating mechanisms to effectively integrate multi-modal data. Building on this, we design a dual-branch causal inference approach: the primary branch focuses on spatio-temporal prediction, while the auxiliary branch mitigates bias by modeling additional modalities and applying causal interventions to uncover true causal dependencies. To improve model efficiency, we integrate GCN with the Mamba architecture for accelerated spatio-temporal encoding. Extensive experiments on 4 real-world datasets show that $E^2$-CSTP significantly outperforms 9 state-of-the-art methods, achieving up to 9.66% improvements in accuracy as well as 17.37%–56.11% reductions in computational overhead.

## 1 Introduction

**Spatio-temporal prediction** plays a critical role in numerous applications, such as intelligent transportation systems [2, 55], weather forecasting [34], environmental monitoring [38], and urban planning [49]. Accurate predictions in these areas help improve decision-making and optimize resource allocation. For example, accurate traffic flow forecasting can improve road safety, while reliable weather predictions facilitate effective disaster preparedness.

Meanwhile, recent advances in information technology have facilitated the proliferation of diverse data types and sources. **Multi-modal data**, comprising information from distinct sensing channels (e.g., satellite images, text, and sensor-based inputs), often exhibits rich cross-modal correlations. Effective integration of such heterogeneous data can mitigate the constraints of single-modal approaches while providing more comprehensive spatio-temporal data representations for enhanced predictive performance [44, 22, 25]. For instance, urban traffic forecasting benefits significantly from incorporating multi-modal inputs like surveillance image and social media text alongside traditional traffic flow data, leading to more accurate predictions [33]. Despite the efforts of previous studies, we observe that **several critical challenges** still persist in multi-modal spatio-temporal prediction.

The first fundamental challenge lies in the insufficient integration of spatio-temporal patterns with heterogeneous multi-modal data. Specifically, prior approaches [42, 12, 8] primarily concentrate on homogeneous modalities within multi-modal datasets. For instance, MoSSL [8] models taxi and bicycle inflow/outflow patterns as distinct modalities for traffic forecasting. However, real-world

---

\*Ziquan Fang is the corresponding author.

39th Conference on Neural Information Processing Systems (NeurIPS 2025).

spatio-temporal systems typically involve deeply interconnected multi-modal data characterized by two key relationships: semantic correlations and complementary information exchange. Fig. 1(a) exemplifies these relationships in traffic prediction scenarios. As observed, (i) temporal traffic flow patterns correlate with aerial image through semantic relationships and (ii) social media text (e.g., road closure reports) provides complementary information to conventional sensor data. Although the state-of-the-art LLM-based studies [48, 20, 30] have explored multi-modal approaches, significant limitations persist. For example, LLM-based text training [48] is modality-specific and lacks generalizability, whereas GPT4MTS [20] and TimeMMD [30] focus solely on temporal patterns without considering spatial dimensions. Besides, several studies [29, 3] employ one modality to predict another. In contrast, the focus of our work lies in utilizing supplementary modalities to enhance spatio-temporal forecasting.

The second challenge is that the causal relations in spatio-temporal prediction are typically confounded by latent variables and environmental biases [70, 52, 45, 13, 63]. Fig. 1(b) illustrates how multi-modal data can give rise to complex causal interactions. As observed, bicycle and taxi flows exhibit a positive correlation on sunny days due to increased outdoor activity, while rainy conditions invert this correlation, suppressing bicycle usage while amplifying taxi demand as commuters seek sheltered transportation. Fig. 1(c)

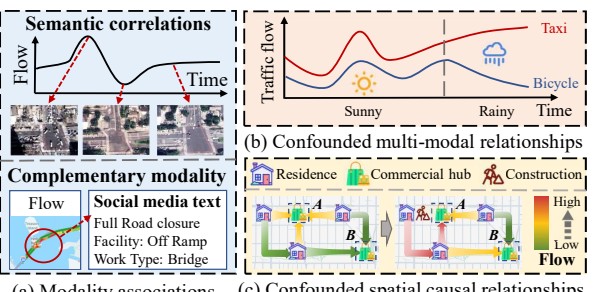

Figure 1: Multi-modal and confounded relations.

further demonstrates a spatially confounded scenario. Commercial hubs like Area A typically attract high traffic volumes, but ongoing construction (a latent confounder) simultaneously reduces road capacity and diverts flow to adjacent Area B. By applying causal inference, true causal relations can be accurately distinguished from confounding factors, thereby improving prediction accuracy.

Moreover, after reviewing prior spatio-temporal prediction methods, we reveal that Transformer-based models [21, 35] have become the popular approach, showing promising performance. However, their quadratic complexity $O(T^2)$ with respect to input spatio-temporal sequence length $T$ imposes severe scalability challenges, as evidenced by several hours of training time for city-scale spatio-temporal prediction in our experiments. While various efficient Transformer variants have been introduced to reduce computational complexity, they often suffer from architectural limitations or degraded modeling capacity [15]. Consequently, the model efficiency bottleneck significantly hinders the practical deployment on large-scale datasets, where both accuracy and computational cost matter.

**Contributions.** To address the above three challenges, we propose $E^2$-CSTP, an **E**ffective and **E**fficient **C**ausal multi-modal **S**patio-**T**emporal **P**rediction framework.

- **Unified Multi-Modal Spatio-Temporal Fusion.** We systematically integrate various modalities (environmental images, event-related text, and spatio-temporal time-series data) through cross-modal attention and adaptive gating mechanisms. To the best of our knowledge, this is the first work to jointly model heterogeneous features in a unified prediction framework, enabling comprehensive representation learning across complementary data sources.

- **Dual-Branch based Causal Disentanglement.** We introduce a dual-branch causal inference design for spatio-temporal prediction. Specifically, the main branch focuses on learning spatio-temporal patterns, while the auxiliary branch models additional modalities and leverages causal interventions to reduce confounding bias from environmental and event-related factors.

- **Efficient and Hybrid Model Design.** We incorporate GCN and Mamba for efficient spatio-temporal encoding. Specifically, GCN captures spatial neighborhood information to reduce the computational load of global dependencies, while Mamba handles temporal dependencies to further decrease computational complexity and accelerate spatio-temporal prediction. This hybrid design achieves faster model inference while maintaining competitive model accuracy.

- **Extensive Experiments.** Through extensive evaluations on four real-world datasets and nine baselines, $E^2$-CSTP achieves up to 9.66% improvement in accuracy, 17.37%–56.11% speedup in model efficiency, and demonstrates consistent robustness across varying parameter settings.

## 2 Related Work

**Single-Modal Spatio-Temporal Prediction.** Early approaches [11, 32, 36] primarily relied on statistical models, which depended on predefined assumptions and the statistical properties of the data. With the advancement of deep learning, models based on Recurrent Neural Networks (RNN) [7, 17, 6], Convolutional Neural Networks (CNN) [41, 51, 60], and Graph Neural Networks (GNN) [61, 66, 18, 56] have shown strong capabilities in modeling temporal, spatial, and structural dependencies, respectively. More detailed single-modal spatio-temporal prediction studies can refer to related surveys [39, 23, 47, 5]. Transformer-based models [43, 28, 21] leverage self-attention for global dependency modeling and excel at capturing long-range temporal correlations. However, their inherent quadratic complexity poses a major limitation for long-term time series forecasting [14], especially in spatio-temporal contexts where attention must model both spatial and temporal dependencies, leading to quadratic scaling with the number of nodes and time steps. Although several Transformer variants [68, 50] reduce complexity through structural simplifications, these modifications may hinder the model's ability to capture complex patterns [15].

**Multi-Modal Spatio-Temporal Prediction.** In time series analysis, multi-modal methods have been applied across domains such as healthcare and finance. For instance, combining heterogeneous data sources (e.g., clinical records, medical images, genomic data) enhances medical predictive accuracy [58], while integrating social and economic signals strengthens market forecasting in finance [46]. Recent work explores LLM-based multi-modal models, GPT4MTS [20] leverages both numerical and textual inputs via prompt-based learning. MM-TSFlib [30] integrates language and time series models through end-to-end training. Wang et al. [48] combine LLMs with generative models for joint reasoning over news events and time series data, improving complex event prediction. However, these approaches focus solely on temporal dynamics and neglect spatial dependencies. In spatio-temporal prediction, Deng et al. [8] adopt self-supervised learning to capture latent patterns in multi-modal spatio-temporal data. Zhang et al. [62] apply AutoML techniques to model the dynamics of multi-modal meteorological data, but their method processes different traffic and weather types separately, with limited cross-modal interaction. Wang [44] utilizes multi-modal data for smart mobility prediction, yet each task relies on a single modality, falling short of unified multi-modal modeling. Zhao et al. [64] incorporate POI and weather data as contextual factors to model multi-modal traffic flow. Yan et al. [53] and Zhou et al. [67] fuse diverse urban data sources, including social media and real estate information, to improve traffic speed prediction. Han et al. [16] employ a pre-trained encoder and multi-modal inputs to model event impacts on traffic. Nonetheless, these models often struggle to model complex cross-modal dependencies and are mostly limited to textual inputs, lacking support for other essential data types like images.

**Spatial-Temporal Causal Inference.** Causal inference aims to uncover cause-effect relations among variables. Integrating causal inference into spatio-temporal forecasting enhances both interpretability and predictive accuracy in complex, dynamic environments. In time series causal discovery, Granger causality uses non-parametric estimation to identify dependencies between variables [1], but assumes full observability of relevant variables. When latent confounders or hidden causal factors are present, causal representation learning becomes necessary. Methods such as iVAE [24], LEAP [54], and GCIM [65] explore temporal causal representations by modeling latent distributions and eliminating spurious correlations. Zhao et al. [63] propose DyGNN Explainer, a dynamic variational graph autoencoder to uncover causal and dynamic relations. CaST [52] and CauSTG [70] address out-of-distribution generalization and dynamic spatial causality via implicit modeling and intervention-based learning. Recent approaches like NuwaDynamics [45] and CaPaint [13] apply causal inference to identify causally relevant regions. However, these works are restricted to single-modal data and do not capture the causal mechanisms underlying multi-modal spatio-temporal interactions.

## 3 Preliminary

The commonly used notations and descriptions are summarized in **Appendix A** (see Table 2).

**Definition 1 (Spatio-Temporal Data).** Spatio-temporal data refers to a sequence of sensor observations collected at discrete time intervals over a spatial graph. Specifically, the spatial graph is denoted as $\mathcal{G} = (\mathcal{V}, \mathcal{E}, \mathbf{A})$, where $\mathcal{V}$ is the set of $N = |\mathcal{V}|$ nodes, $\mathcal{E}$ is the edge set, and $\mathbf{A} \in \mathbb{R}^{N \times N}$ is the adjacency matrix encoding pairwise spatial relationships. The spatio-temporal data is denoted as $X = [x^{t-T+1}, \ldots, x^t] \in \mathbb{R}^{T \times N \times d}$, where $T$ is the number of historical time steps, $N$ is the number of spatial nodes, and $d$ is the feature dimension at each node.

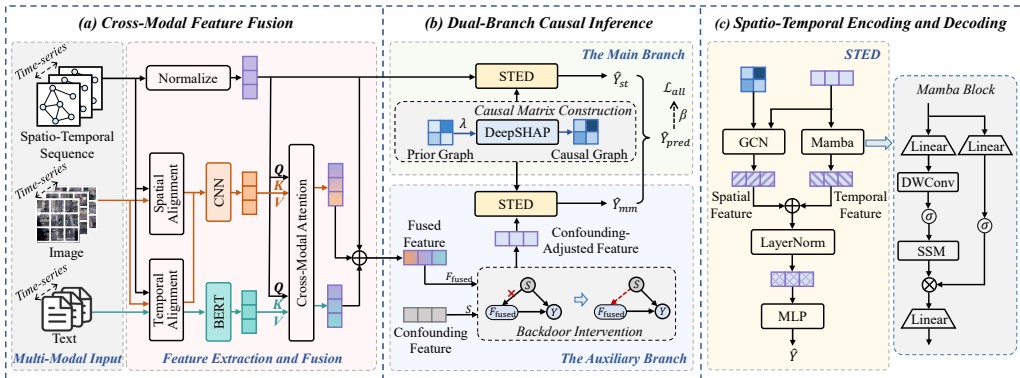

Figure 2: The overall framework of E$^2$-CSTP.

**Definition 2 (Multi-Modal Spatio-Temporal Data).** Multi-modal spatio-temporal data extends spatio-temporal signals by incorporating heterogeneous sources of information across different modalities, such as spatio-temporal sequence, event-related text, or visual images. Let $\mathcal{X} = \{X_1, X_2, \ldots, X_n\}$ represent the $n$ modalities, where each modality consists of spatio-temporal observations defined as $X_i = [x_i^{t-T_i+1}, \ldots, x_i^t] \in \mathbb{R}^{T_i \times N_i \times d_i}$. Here, $T_i$ denotes the number of time steps (i.e., the temporal resolution), $N_i$ is the number of spatial units (i.e., the spatial resolution), and $d_i$ represents the feature dimension of the $i$-th modality.

**Problem Statement (Multi-Modal Spatio-Temporal Prediction).** Given the graph $\mathcal{G}$ and the historical $T$-step of multi-modal spatio-temporal data $\mathcal{X}$, where $\mathcal{X}$ may include spatio-temporal sequence and auxiliary modalities, we aim to learn a function $\theta(\cdot)$ that leverages $\mathcal{X}$ to predict the future $S$-step spatio-temporal data $Y_{\text{st}} = [y_{\text{st}}^{t+1}, \ldots, y_{\text{st}}^{t+S}] \in \mathbb{R}^{S \times N_{st} \times d}$.

## 4 Methodology

**Framework Overview.** As illustrated in Fig. 2, the E$^2$-CSTP framework consists of three key components: (i) cross-modal feature fusion, (ii) dual-branch causal inference, and (iii) spatio-temporal encoding and decoding (STED). These modules work in concert to integrate multi-modal signals and apply causal inference techniques to reduce the impact of external confounders, thereby enhancing both prediction accuracy and model interpretability. Next, we detail each module below.

### 4.1 Cross-Modal Feature Fusion

We propose a cross-modal attention mechanism combined with a fusion gating module to integrate spatio-temporal, textual, and visual signals into a unified latent representation. Unlike naive concatenation or static fusion approaches, our method dynamically attends to relevant modality-specific features. The fusion process proceeds through the following steps.

**Step 1: Multi-modal feature extraction and alignment.** Multi-modal data typically exhibits substantial differences in temporal resolution and semantic content. To address this issue, we first perform feature alignment operations across different modalities, as shown in Fig. 2(a).

For text data, which lacks spatial information, alignment is performed solely along the temporal dimension. Event timestamps $\tau$ are extracted using regular expression parsing, and each text timestamp is matched to its nearest spatio-temporal point in time. For image data, alignment incorporates both temporal and spatial dimensions. Each image is associated with a timestamp $\tau$ and spatial coordinates $\rho$, and is aligned to the nearest spatio-temporal point based on combined temporal and spatial proximity. We define the unified binary alignment matrix as follows:

$$\mathbf{M}_{i,j}^t = \begin{cases} 1, & \text{if } j = \arg\min_k |\tau^{(i)} - \tau_{\text{st}}^{(k)}| \\ 0, & \text{otherwise} \end{cases}, \mathbf{M}_{i,j}^s = \begin{cases} 1, & \text{if } j = \arg\min_k \|\rho^{(i)} - \rho_{\text{st}}^{(k)}\|_2 \\ 0, & \text{otherwise} \end{cases}, \quad (1)$$

where $\mathbf{M}_{i,j}^t$ and $\mathbf{M}_{i,j}^s \in \{0,1\}^{T \times T_{\text{st}}}$ denote the temporal and spatial alignment matrices, respectively. $\tau^{(i)}$ represents the timestamp of the $i$-th text or image observation, and $\tau_{\text{st}}^{(k)}$ denotes the timestamp of

the $k$-th element in the spatio-temporal sequence. Similarly, $\rho^{(i)}$ and $\rho^{(k)}_{\text{st}}$ indicate the corresponding spatial coordinates of the $i$-th observation and the $k$-th spatio-temporal point, respectively.

Subsequently, the aligned features are computed by performing matrix multiplication independently across each feature channel $d$. The text features are then duplicated $N_{st}$ times to match the target dimensionality, resulting in the following expression, as shown below:

$$\widetilde{X}_{\text{text}} = \text{repeat}(\mathbf{M}^{t^\top} X_{\text{text}}, N_{\text{st}}), \quad \widetilde{X}_{\text{img}} = \mathbf{M}^{t^\top} X \mathbf{M}^s, \tag{2}$$

where $\widetilde{X}_{\text{text}} \in \mathbb{R}^{T_{\text{st}} \times N_{\text{st}} \times d_{\text{text}}}$ and $\widetilde{X}_{\text{img}} \in \mathbb{R}^{T_{\text{st}} \times N_{\text{st}} \times d_{\text{img}}}$.

Finally, the spatio-temporal sequence $X_{\text{st}}$ is normalized to $F_{\text{st}}$ to ensure consistent scaling across time steps. To extract semantic representations from unstructured text, we employ a pre-trained BERT model[9] to encode textual inputs into contextualized embeddings aligned with the spatio-temporal context, represented as $F_{\text{text}} = \text{BERT}(\widetilde{X}_{\text{text}})$. Simultaneously, visual features are extracted from images using a convolutional neural network (CNN), formulated as $F_{\text{img}} = \text{CNN}(\widetilde{X}_{\text{img}})$, which highlights geographic cues relevant to environmental conditions.

**Step 2: Multi-modal feature fusion.** To facilitate interactions among features from different modalities within a shared latent space, we project the **spatio-temporal features** $F_{\text{st}}$, **text features** $F_{\text{text}}$, and **image features** $F_{\text{img}}$ into a unified hidden dimension $d$ using three separate fully connected layers. This yields modality-specific representations $\widetilde{F}_{\text{st}}$, $\widetilde{F}_{\text{text}}$, and $\widetilde{F}_{\text{img}} \in \mathbb{R}^{T_{\text{st}} \times N_{\text{st}} \times d}$.

Next, we employ two Cross-Modal Attention (CMA) modules to capture interactions between spatio-temporal, text and image features. Each CMA module computes queries ($Q$), keys ($K$), and values ($V$) from the aligned features and applies a scaled dot-product attention mechanism to model cross-modal similarity. The attention operation is defined as:

$$Attn_{\text{st}\to\text{text}} = \text{CMA}(\widetilde{F}_{\text{st}}, \widetilde{F}_{\text{text}}, \widetilde{F}_{\text{text}}), \quad Attn_{\text{st}\to\text{img}} = \text{CMA}(\widetilde{F}_{\text{st}}, \widetilde{F}_{\text{img}}, \widetilde{F}_{\text{img}}), \tag{3}$$

where $\text{CMA}(Q, K, V) = \text{softmax}(\frac{QK^\top}{\sqrt{d_k}})V$ and $d_k$ represents the dimension of each attention head.

This attention mechanism facilitates information exchange across modalities, thereby enriching the feature representations. To integrate the outputs, we concatenate the modality-specific features and apply a fusion gating mechanism to get the gating values, as shown below:

$$F_{\text{fused}} = [\widetilde{F}_{\text{st}}, Attn_{\text{st}\to\text{text}}, Attn_{\text{st}\to\text{img}}] \odot \text{FusionGate}([\widetilde{F}_{\text{st}}, Attn_{\text{st}\to\text{text}}, Attn_{\text{st}\to\text{img}}]) \tag{4}$$

The gating values regulate the contribution of each modality, producing the final fused representation.

### 4.2 Dual-Branch Causal Inference

Given that multi-modal data introduces complex causal interactions that can impact prediction accuracy, we propose an innovative dual-branch causal invariance approach to differentiate true causal relations from confounding factors, which is shown in Fig. 2(b).

**Step 1: Causal matrix construction.** To uncover latent dependencies among spatial units and enhance the interpretability of the model, we estimate the underlying causal structure using DeepSHAP-based feature importance. The SHAP value, $\phi_{i,j}$, quantifies the influence of node $i$ on node $j$, resulting in a matrix $\mathbf{A}^{\text{SHAP}} \in \mathbb{R}^{N \times N}$. If a prior graph $\mathbf{A}^{(0)}$ is available, we construct a hybrid adjacency matrix, as shown below:

$$\mathbf{A} = \lambda \mathbf{A}^{(0)} + (1 - \lambda)\mathbf{A}^{\text{SHAP}}, \tag{5}$$

where $\lambda \in [0, 1]$ balances reliance on prior knowledge versus data-driven insights.

To enhance model stability and robustness, we adopt an exponential moving average and update the adjacency matrix every $P = 5$ epochs. This interval, determined empirically, balances stability and computational efficiency. Updating too frequently introduces noisy structural fluctuations, while a moderate update interval allows the model to refine its internal representations and capture meaningful spatio-temporal dependencies more effectively.

**Step 2: Dual-branch causal adjustment.** Spatio-temporal prediction is often biased by unobserved confounders ($S$) as well as observed external factors, including image features ($E$, derived from

$Attn_{\text{st}\rightarrow\text{img}}$) and text features ($C$, derived from $Attn_{\text{st}\rightarrow\text{text}}$). To mitigate these biases, we introduce a dual-branch causal adjustment mechanism that explicitly accounts for confounding influences.

The main branch relies solely on the spatio-temporal sequence $X_{\text{st}}$. However, this branch may overlook critical information from external factors. The auxiliary branch integrates multi-modal data $F_{\text{fused}}$ to predict the future sequence. We use a multi-layer perceptron (MLP) layer to combine the output features from the two branches. The final prediction and corresponding loss functions are:

$$\hat{Y}_{\text{final}} = \text{MLP}(f(X_{\text{st}}, \mathbf{A}); f(F_{\text{fused}}, \mathbf{A})) \tag{6}$$

$$\mathcal{L}_{\text{st}} = \|Y_{\text{st}} - f(X_{\text{st}}, \mathbf{A})\|_2, \quad \mathcal{L}_{\text{mm}} = \|Y_{\text{st}} - f(F_{\text{fused}}, \mathbf{A})\|_2, \quad \mathcal{L}_{\text{pred}} = \|Y_{\text{st}} - \hat{Y}_{\text{final}}\|_2, \tag{7}$$

where $f(\cdot)$ denotes the prediction module STED (to be detailed in Section 4.3), $\mathcal{L}_{\text{st}}$ is the loss of the main branch, $\mathcal{L}_{\text{mm}}$ is the loss of the auxiliary branch, and $\mathcal{L}_{\text{pred}}$ is the loss of the final prediction.

Overall, we aim to obtain a model by training on the two branches described above, which is handled by the following loss function.

$$\mathcal{L}_{\text{all}} = \mathcal{L}_{\text{pred}} + \beta\mathcal{L}_{\text{st}} + (1 - \beta)\mathcal{L}_{\text{mm}}, \tag{8}$$

where $\beta$ is an adjusting parameter to balance the influence of the two branches.

Note that although the multi-modal fusion $F_{\text{fused}}$ enriches the representation, concatenating the visual feature $E$ and the textual feature $C$ can amplify the influence of an unobserved confounder $S$, which simultaneously affects both the spatio-temporal signal $X_{\text{st}}$ and the target $Y_{\text{st}}$. We formalize the data generation process using the following Structural Causal Model (SCM), as shown below:

$$\begin{cases} X_{\text{st}} = f_X(S, E, C) \\ Y_{\text{st}} = f_Y(X_{\text{st}}, S, E, C) \end{cases}, \tag{9}$$

where $S$ denotes the confounding feature, approximated by a set of latent variables $\{s_i\}$.

Under the backdoor criterion [37], we estimate interventional outcomes, as shown below:

$$P(Y_{\text{st}}|do(X_{\text{st}} = x), E, C) = \int_S P(Y_{\text{st}}|X_{\text{st}} = x, S = s_i, E, C)P(S = s_i|E, C)dS \tag{10}$$

The intervention on each $x$ is adjusted as follows:

$$\hat{x} = x + x \odot W[\alpha_1 h(S) + \alpha_2 p(E) + \alpha_3 q(C)], \tag{11}$$

where $\hat{x}$ is the intervention result on $x$, $W$ is a set of weights that control the magnitude of influence of the respective factors, and $h(\cdot)$, $p(\cdot)$, and $q(\cdot)$ are functions representing $S$, $E$, and $C$, respectively.

To ensure that the adjustment effectively removes the confounding influence of $S$, we aim to make $\hat{x}$ statistically independent of $S$, conditioned on $E$ and $C$. This can be achieved by minimizing the gradient of $\hat{x}$ with respect to $S$. The training objective is to minimize the influence of the latent variable $S$ on $\hat{x}$, thereby driving $\frac{\partial \hat{x}}{\partial S} \rightarrow 0$. Consequently, the backdoor path $X_{\text{st}} \leftarrow S \rightarrow Y_{\text{st}}$ is blocked, ensuring the validity of the causal inference process. (See **Appendix B** for detailed proof).

### 4.3 Spatio-Temporal Encoding and Decoding (STED)

To capture both spatial and temporal dependencies, we design a spatio-temporal encoding module that combines Graph Convolutional Networks (GCN) and the Mamba model. As shown in Fig. 2(c), the GCN captures spatial neighborhood relations, while Mamba learns temporal evolution features.

The spatial encoding is performed via multi-layer Graph Convolutional operations on node features $X \in \mathbb{R}^{B \times T \times N \times d}$, where $B$ is the batch size, $T$ is the historical time step, $N$ is the number of nodes and $d$ is the feature dimension. The message-passing process is performed using the edge indices constructed from the adjacency matrix $\mathbf{A} \in \mathbb{R}^{N \times N}$, which encodes the spatial relationships between the nodes. The spatial encoding is computed as:

$$X_{\text{spatial}} = \text{GCN}(X, \mathbf{A}) \tag{12}$$

For temporal modeling, while several efficient Transformer variants have been proposed to reduce complexity, they often impose architectural constraints or lose modeling fidelity [15]. In contrast, we employ Mamba[14], a selective state-space model that maintains linear complexity while effectively

capturing long-range dependencies, akin to attention mechanisms. Specifically, each Mamba block consists of an input projection, depth-wise convolution, a gated selective state-space kernel, element-wise modulation, and an output projection (Fig. 2(c), right).

$$X_{\text{temporal}} = \text{Mamba}(X \cdot W_{\text{in}})W_{\text{out}}, \tag{13}$$

where $W_{\text{in}}$ and $W_{\text{out}}$ are the linear projection layers.

Spatial and temporal features are then fused element-wise operations. To enhance training stability, residual connections are incorporated during the fusion process. Each fused output is subsequently passed through a LayerNorm operation to normalize the features, alleviating internal covariate shift and facilitating the training of deeper models.

$$X_{\text{encoded}} = \text{LayerNorm}(X_{\text{spatial}} + X_{\text{temporal}}) \tag{14}$$

The module consists of three stacked layers of GCN and Mamba substructures, alternating between capturing spatial and temporal dependencies. The final output $X_{\text{encoded}}$ is a set of spatio-temporal encoded features that match the shape of the input.

Finally, the spatio-temporal encoded features are decoded through an MLP layer to generate the output predictions. The decoding process is expressed as follows:

$$\hat{Y} = \text{MLP}(X_{\text{encoded}}) \tag{15}$$

### 4.4 Model Training

**Training Algorithm.** The overall training algorithm of E$^2$-CSTP is described in **Appendix C**. Compared to Transformer-based methods with quadratic complexity, our hybrid GCN-Mamba architecture significantly reduces computational overhead while maintaining predictive performance.

**Complexity Analysis.** In standard Transformer architectures, the input tensor $X \in \mathbb{R}^{B \times T \times N \times d}$ is typically flattened along the temporal and spatial dimensions, resulting in a sequence of length $S = T \cdot N$. The self-attention mechanism incurs a time complexity of $O(B \cdot T^2 \cdot N^2 \cdot d)$. This quadratic dependency on both the number of nodes and time steps significantly increases computational cost, particularly when dealing with long sequences or large spatial graphs.

In contrast, the proposed STED module decomposes spatio-temporal modeling into spatial GCN and temporal Mamba components, each operating along a single axis. The time complexity of the GCN is $O(B \cdot T \cdot N^2 \cdot d)$. The Mamba block operates per node and time step, with linear-time complexity in $T$. Each operation (projection, convolution, kernel application, modulation) is applied to a sequence of length $T$ for each of the $N$ nodes and $B$ samples and the complexity is $O(B \cdot T \cdot N \cdot d)$. Therefore, the overall complexity is $O(B \cdot T \cdot N^2 \cdot d)$, enabling faster and more memory-efficient training on large-scale spatio-temporal data.

In addition, the proposed STED module, whose complexity scales with sequence length $T$ and graph size $N$, drives the speed-up, while the shared text and image encoders merely handle preprocessing and therefore do not affect the relative ranking.

## 5 Experiments

We use below four Research Questions (RQs) to guide the experiments.

***RQ1.*** *How does E$^2$-CSTP perform compared to existing single-modal and multi-modal spatio-temporal prediction methods?*

***RQ2.*** *How do the various modules in E$^2$-CSTP contribute to its performance?*

***RQ3.*** *How efficient is E$^2$-CSTP in training compared to both baseline models and Transformer-based prediction module alternatives?*

***RQ4.*** *How sensitive is E$^2$-CSTP to changes in hyperparameter settings?*

### 5.1 Experimental Settings

**Datasets.** We collect 4 datasets to evaluate the proposed E$^2$-CSTP framework: Terra [4], BjTT [59], GreenEarthNet [3], and BikeNYC [60]. (i) Terra provides spatio-temporal observations along with multi-modal information such as geo-images and explanatory texts; (ii) BjTT is a multi-modal

Table 1: Overall performance. Best results are **bold** and the second best are underlined.

| Method | Terra | | | BjTT | | | GreenEarthNet | | | BikeNYC | | |
|---|---|---|---|---|---|---|---|---|---|---|---|---|
| | MAE | RMSE | MAPE | MAE | RMSE | MAPE | MAE | RMSE | MAPE | MAE | RMSE | MAPE |
| D2STGNN | 2.52 | 3.13 | 29.79% | 4.57 | 7.75 | 14.47% | 0.22 | 0.28 | 79.10% | 6.10 | 10.55 | 69.06% |
| ST-SSL | 2.55 | 3.18 | 30.32% | 4.83 | 8.00 | 15.78% | 0.24 | 0.31 | 84.45% | 7.68 | 14.02 | 78.93% |
| HimNet | 2.54 | 3.15 | 26.83% | 3.79 | 6.10 | 11.02% | 0.16 | 0.21 | 76.33% | 4.62 | 7.93 | 64.58% |
| NuwaDynamics | 2.49 | 3.07 | 24.85% | 3.72 | 5.98 | 10.79% | 0.18 | 0.26 | 73.26% | 3.56 | 6.27 | 61.68% |
| CaPaint | 2.49 | 3.08 | 25.47% | 3.74 | 6.03 | 10.89% | 0.18 | 0.25 | 68.37% | 3.54 | 6.10 | 62.95% |
| GPT-ST | 2.47 | 3.06 | 30.06% | 3.69 | 5.66 | 10.02% | 0.17 | 0.23 | 98.83% | 3.37 | 5.85 | 63.97% |
| UniST | 2.47 | 3.05 | 25.02% | 3.62 | 5.49 | 9.62% | 0.14 | 0.20 | 72.79% | 3.31 | 5.58 | 60.58% |
| T3 | 2.53 | 3.11 | 26.13% | 3.73 | 6.02 | 11.01% | - | - | - | - | - | - |
| FNF | 2.51 | 3.10 | 27.42% | 3.64 | 5.51 | 9.73% | - | - | - | - | - | - |
| E$^2$-CSTP | **2.43** | **3.01** | **23.62%** | **3.56** | **5.32** | **9.24%** | **0.13** | **0.18** | **57.09%** | **2.99** | **5.53** | **56.13%** |

dataset containing traffic data and event descriptions for traffic prediction; (iii) GreenEarthNet is a multi-modal satellite dataset used to estimate vegetation; (iv) BikeNYC contains only spatio-temporal sequences, evaluating model performance based on the bike flow attribute in a single-modality setting. The datasets are chronologically divided into training, validation, and test sets in an 8:1:1 ratio. We use data from the past 12 time steps to predict the subsequent 12 time steps. More detailed dataset information is provided in **Appendix D.1**.

**Baselines.** We compare E$^2$-CSTP with 9 state-of-the-art spatio-temporal prediction methods, categorized into four groups: (i) **Single-modal** spatio-temporal prediction methods, including D2STGNN [40], ST-SSL [19], and HimNet [10]; (ii) **Causality-based** spatio-temporal prediction methods, including NuwaDynamics [45] and CaPaint [13]; (iii) **Foundation models** for spatio-temporal prediction, including GPT-ST [27] and UniST [57]; (iv) **Multi-modal** spatio-temporal prediction methods, including T3 [16] and From News to Forecast (FNF) [48]. For more detailed information of the baselines and implementation, please refer to **Appendix D.2** and **Appendix D.3**.

**Evaluation Metrics.** We evaluate model performance using Mean Absolute Error (MAE), Root Mean Square Error (RMSE), and Mean Absolute Percentage Error (MAPE) for accuracy, and total runtime and per-epoch runtime for efficiency. Detailed calculation methods are provided in **Appendix D.4**.

## 5.2 Overall Performance Comparison (*RQ1*)

Table 1 presents the overall performance comparison of all methods across the four datasets. We yield the following observations. Notably, T3 and FNF are excluded from comparisons on GreenEarthNet and BikeNYC due to the absence of textual modalities in these datasets.

First, E$^2$-CSTP consistently outperforms all baselines across metrics and datasets, with MAE improvements ranging from **1.61% to 9.66%** over the second-best methods. This highlights the effectiveness of our E$^2$-CSTP framework in capturing fine-grained dependencies often missed by other methods.

Second, multi-modal and causality-based models generally outperform single-modal baselines, confirming the necessity of auxiliary modalities and causal reasoning in spatio-temporal forecasting. However, causality-based models, despite their theoretical appeal, often lack the contextual understanding required for comprehensive predictions. In contrast, E$^2$-CSTP integrates causal reasoning within a multi-modal framework, further reducing uncertainty through cross-modal interactions.

Third, E$^2$-CSTP consistently outperforms other multi-modal methods on environment- and event-driven datasets, with up to 3.95% performance gain. This advantage stems from its ability to suppress confounding factors and leverage rich visual and textual cues, thereby enabling robust predictions in complex real-world scenarios. In contrast, foundation models such as GPT-ST and UniST, despite their strong transferability from large-scale pre-training, lack explicit multi-modal fusion and causal modeling, which limits their fine-grained predictive capabilities.

## 5.3 Ablation Study (*RQ2*)

Next, we conduct ablation studies to assess the contribution of six components in our E$^2$-CSTP framework. **(1) w/o Text Feature:** It removes text inputs to assess the impact of language-based auxiliary information. **(2) w/o Image Feature:** It excludes visual inputs to evaluate the contribution of image context. **(3) w/o DeepSHAP:** It uses only the initial adjacency matrix, omitting the DeepSHAP-based causal region identification. **(4) w/o Causal Inference (CI):** It disables the causal intervention to examine the importance of confounder mitigation. **(5) w/o GCN:** It removes the spatial encoding,

disabling the graph-based spatial dependency modeling. **(6) w/o Mamba:** It eliminates temporal encoding based on the Mamba architecture, testing its contribution to temporal dynamics modeling.

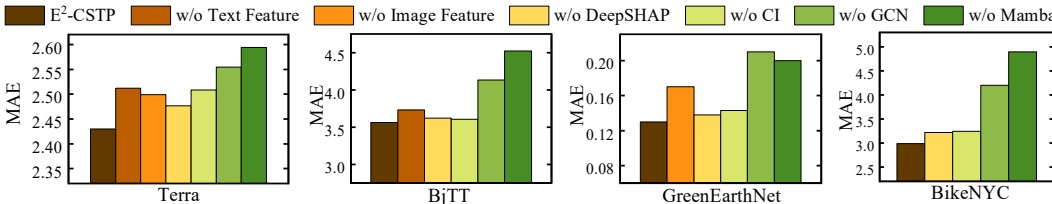

Figure 3: The ablation study.

The results are shown in Fig. 3. The w/o Text Feature and w/o Image Feature variants suffer performance drops, indicating that both textual and visual inputs provide essential contextual cues for accurate prediction. The w/o DeepSHAP model fails to effectively focus on influential regions, weakening spatial reasoning. Removing the Causal Inference module leads to reduced robustness, especially under confounding conditions. The w/o GCN variant struggles with spatial structure modeling, while the w/o Mamba variant cannot capture temporal dynamics effectively. These results confirm that every module is critical for modeling complex multi-model spatio-temporal patterns.

### 5.4 Model Efficiency Study (*RQ3*)

We further evaluate the efficiency of $E^2$-CSTP by comparing it with 9 spatio-temporal baseline models. Fig. 4 shows the total training time required to reach convergence under the same batch size.

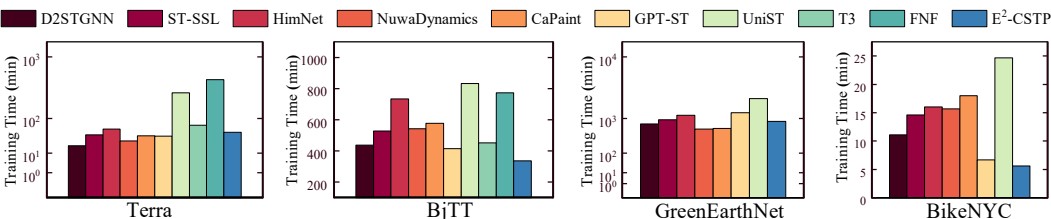

Figure 4: Model efficiency comparison on the total training time.

The results indicate that $E^2$-CSTP achieves training time comparable to those of single-modal methods, even when applied to multi-modal datasets. Moreover, it consistently outperforms UniST and other multi-modal models, showing a particularly notable advantage over LLM-based approaches such as FNF. On the single-modal BikeNYC dataset, $E^2$-CSTP achieves faster training compared to all baseline methods, further showing its superiority.

To assess the effectiveness of our prediction module STED on computational cost, we replace it with several popular Transformer-based alternatives, including Informer [68], Autoformer [50], FED-former [69], and iTransformer [31], thus isolating the gain brought by replacing a quadratic Transformer with our linear GCN-Mamba block. These variants are denoted as w/ In, w/ Auto, w/ FED, and w/ iTrans, respectively. Fig. 5 shows the prediction accuracy and per-epoch runtime on the Terra dataset. As observed, our method improves prediction accuracy by 1.78%–5.45% while reducing computational overhead by **17.37%–56.11%** compared to these alternatives, demonstrating that the proposed module is both effective and efficient in practice.

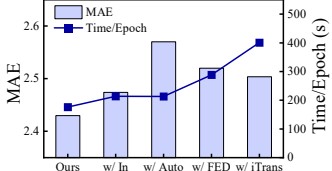

Figure 5: Efficiency under prediction variants on Terra.

Due to space limitations, detailed per-epoch runtime comparisons with 9 baseline models and additional results from alternative Transformer-based prediction modules on other datasets are provided in **Appendix D.5**, showing similar observations.

### 5.5 Parameter Sensitivity Study (*RQ4*)

To evaluate the impact of hyperparameters, we vary the graph fusion factor $\lambda$ across $\{0, 0.25, 0.5, 0.75, 1\}$ and the dual-branch loss balancing factor $\beta$ across $\{0, 0.25, 0.5, 0.75, 1\}$, selecting the optimal combination for each dataset based on validation performance. Specifically, for the Terra and GreenEarthNet datasets, $\lambda$ is set to 0.25, with $\beta$ set to 0.75 and 0.5, respectively. For

the BjTT and BikeBYC datasets, $\lambda$ is set to 0.5, while $\beta$ is set to 0.5 and 0.75, respectively. Detailed parameter analysis can be found in **Appendix D.6**.

## 6 Conclusion

In this paper, we propose $E^2$-CSTP for spatio-temporal prediction that addresses the challenges of multi-modal fusion, confounding bias, and computational inefficiency. By integrating cross-modal attention and gating mechanisms, $E^2$-CSTP achieves robust fusion of spatio-temporal, textual, and visual inputs. To mitigate biases introduced by auxiliary inputs, we introduce dual-branch causal inference based on causal interventions. Experiments conducted on 4 real datasets demonstrate that $E^2$-CSTP outperforms 9 SOTA baselines in accuracy and efficiency. A more detailed discussion of the limitations and potential future directions is provided in **Appendix E**.

## 7 Acknowledgment

This work was supported in part by the NSFC under Grants No. (62402422, 62025206, U23A20296, and 62472377), Yongjiang Talent Introduction Programme (2024A-162-G), Zhejiang Provincial Natural Science Foundation of China under Grant No. LZ25F020001, and Zhejiang Province's "Lingyan" R&D Project under Grant No. 2024C01259. Ziquan Fang is the corresponding author.

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

# Appendix

## A  Notations and Descriptions

Table 2 presents the frequently used notations and their descriptions.

Table 2: Notations and descriptions

| Notation | Description |
|---|---|
| $\mathcal{V}, N$ | Set of nodes and number of nodes in the spatial graph |
| $\mathcal{E}, \mathbf{A}$ | Set of edges and adjacency matrix of the spatial graph |
| $\mathcal{G}$ | Spatial graph $\mathcal{G} = (\mathcal{V}, \mathcal{E}, \mathbf{A})$ |
| $X$ | Multi-modal spatio-temporal data $\{X_{\text{st}}, X_{\text{text}}, X_{\text{img}}\}$ |
| $Y, \hat{Y}$ | Future and predicted spatio-temporal sequence |
| $\tau$ | Timestamps of spatio-temporal, text, and image data |
| $\rho$ | Spatial coordinates for image and spatio-temporal data |
| $\mathbf{M}$ | Alignment matrices for text and image modalities |
| $F$ | Features from spatio-temporal, text, and image modalities |
| $S, E, C$ | Latent confounder, image, and text variables |

## B  Causal inference

### B.1  The definition of causal inference

Causal inference seeks to quantify how interventions influence outcomes of interest. A Structural Causal Model (SCM) specifies the underlying data-generating mechanism. The do-operator supplies the formal notation for interventions. Identifiability criteria, most notably the backdoor and front-door rules, convert causal queries into estimable statistical functionals. Pearl's [37] three rules of do-calculus unify these components, allowing one to strip away interventions iteratively whenever the requisite d-separation conditions are met.

### B.2  The three rules of *do*-calculus

Pearl's *do*-calculus provides symbolic rules that transform interventional distributions to observational ones by exploiting graphical separation statements. Let $G_{do(x)}$ be the graph obtained from SCM after performing $do(X = x)$. $G_{\text{null}(z)}$ denotes the graph where incoming edges to $Z$ are cut but $Z$ is not fixed to a constant.

*Rule 1 (Insertion/deletion of observations).*

$$P(y \mid do(x), z) = P(y \mid do(x)) \quad \text{if } y \perp\!\!\!\perp z \mid x \text{ in } G_{do(x)} \tag{16}$$

*Rule 2 (Action/observation exchange).*

$$P(y \mid do(x), do(z)) = P(y \mid do(x), z) \quad \text{if } y \perp\!\!\!\perp z \mid x \text{ in } G_{do(x), \text{null}(z)} \tag{17}$$

*Rule 3 (Insertion/deletion of actions).*

$$P(y \mid do(x), do(z)) = P(y \mid do(x)) \quad \text{if } y \perp\!\!\!\perp z \mid x \text{ in } G_{do(x), do(z)} \tag{18}$$

These rules are complete for transforming interventional distributions into observational ones and underpin backdoor adjustment.

### B.3  Deriving the backdoor adjustment

Consider a treatment–outcome pair $(X, Y)$ and a set of variables $Z$ such that (i) $Z$ blocks every backdoor path from $X$ to $Y$ in $\mathcal{G}$, and (ii) $Z$ contains no descendant of $X$. We recover the celebrated backdoor formula by successive applications of the do-calculus rules.

*Step 1 (Start from Bayes' rule).*

$$P(y \mid do(x)) = \sum_z P(y \mid do(x), z) \, P(z \mid do(x)) \tag{19}$$

*Step 2 (Apply Rule 3 to remove the action on $Z$).*

$$P(y \mid do(x), z) = P(y \mid x, z) \tag{20}$$

*Step 3 (Substitute the two observations into Step 1).*

$$P(y \mid do(x)) = \sum_z P(y \mid x, z) \, P(z), \tag{21}$$

which is the backdoor adjustment formula. When $Z$ is continuous, the summation in Eq. 21 is replaced by an integral.

## B.4 Validity of the adjustment mechanism

Eq. 21 states that, whenever a valid back-door set $Z$ exists, causal effects can be estimated by (i) stratifying on $Z$, (ii) computing the conditional association between $X$ and $Y$ within each stratum, and (iii) averaging the results over the marginal distribution of $Z$. This principle underlies the dual-branch causal adjustment in **Section 4.2** of the main paper, where the latent confounder $S$ plays the role of $Z$.

*Proof.* We formally justify that the adjusted representation $\hat{x}$ in **Section 4.2** satisfies two essential conditions:

$$\hat{x} \sim P(X_{\text{st}} \mid do(X_{\text{st}}), E, C) \quad \text{and} \quad \hat{x} \perp\!\!\!\perp S \mid E, C, \tag{22}$$

ensuring structural disentanglement between environmental encoding $E$ and event encoding $C$ without mutual interference.

*(1) Cross-modality independence.*

We assert that the learned representations of environmental $E$ and event $C$ variables are distributionally independent:

$$p(E) \perp\!\!\!\perp q(C), \tag{23}$$

and their respective gradients with respect to each other vanish:

$$\frac{\partial p}{\partial C} = 0, \quad \frac{\partial q}{\partial E} = 0, \tag{24}$$

indicating disentangled representations without cross-modal entanglement.

*(2) Conditional independence of $\hat{x}$ and $S$.*

To eliminate the influence of confounding variable $S$ on the adjusted representation $\hat{x}$, we define an optimization objective to minimize the conditional variance of $\hat{x}$ given $E$ and $C$, with respect to $S$:

$$\min_{\alpha_1, \alpha_2, \alpha_3} \mathbb{E}\left[ (\hat{x} - \mathbb{E}[\hat{x} \mid E, C])^2 \mid S \right] \tag{25}$$

The adjusted representation $\hat{x}$ is formulated as:

$$\hat{x} = x + x \odot \left( \alpha_1 h(S) + \alpha_2 p(E) + \alpha_3 q(C) \right), \tag{26}$$

where $\odot$ denotes element-wise multiplication, and $\alpha_1, \alpha_2, \alpha_3$ are gating coefficients modulating the influence of confounder, environment, and event, respectively.

During adversarial training, the confounder-invariant component is encouraged via gradient cancellation:

$$x \odot W \alpha_1 \frac{\partial h}{\partial S} \approx -\frac{\partial x}{\partial S}(1 + W \alpha_1 h(S)) \tag{27}$$

This leads to a representation of the form:

$$\hat{x} \approx f_X(S_0, E, C) + x \odot W \left[ \alpha_2 p(E) + \alpha_3 q(C) \right], \tag{28}$$

where $S_0$ represents a fixed baseline value of the confounder. Consequently, $\hat{x}$ becomes insensitive to $S$ variations and approximates sampling from an interventional distribution conditioned on $E$ and $C$ under a fixed $S$, approximating $P(X_{\text{st}} \mid do(X_{\text{st}}), E, C)$.

In conclusion, $\hat{x}$ fulfills conditions necessary for reliable causal inference in multi-modal fusion models. It maintains independence from confounding factors and preserves the disentanglement between environmental and event encodings. Thus, $\hat{x}$ provides a valid approximation of $P(X_{\text{st}} \mid do(X_{\text{st}}), E, C)$, crucial for interpretable and robust causal analysis in complex data environments.

## C  Training

The detailed algorithmic procedure for our framework can be found in Algorithm 1.

In the preprocessing stage (line 1), we align text and image data with the spatio-temporal sequence to obtain $\widetilde{X}_{\text{text}}$ and $\widetilde{X}_{\text{img}}$, followed by feature extraction via BERT and CNN (line 2-3). We then compute cross-modal attention between $F_{\text{st}}$ and the auxiliary modalities (lines 4–7), and apply a gating-based fusion mechanism to obtain $F_{\text{fused}}$ (line 8-9). An adjacency matrix $\mathbf{A}$ is constructed by combining prior knowledge and SHAP-based feature importance (line 10-11).

The prediction function STED (lines 12–17) models spatial dependencies via GCN and temporal dynamics via Mamba, and outputs the predictions via MLP. During training (lines 18–25), we generate predictions from both $X_{\text{st}}$ and backdoor-adjusted fused features. Corresponding losses $\mathcal{L}_{\text{pred}}$, $\mathcal{L}_{\text{st}}$, and $\mathcal{L}_{\text{mm}}$ are computed, and the model parameters are updated by minimizing the overall loss $\mathcal{L}_{\text{all}}$.

---

**Algorithm 1:** Training procedure of E$^2$-CSTP

**Input:** Spatio-temporal data $X_{\text{st}}$, text data $X_{\text{text}}$, image data $X_{\text{img}}$

**Output:** Predicted spatio-temporal sequence $\hat{Y}$

1 ▷ Preprocessing: Align modalities to obtain $\widetilde{X}_{\text{text}}$, $\widetilde{X}_{\text{img}}$ aligned with $X_{\text{st}}$;
2 ▷ Feature Extraction and Normalization:
3 $F_{\text{text}} \leftarrow \text{BERT}(\widetilde{X}_{\text{text}})$; $F_{\text{img}} \leftarrow \text{CNN}(\widetilde{X}_{\text{img}})$; $F_{\text{st}} \leftarrow \text{Normalize}(\widetilde{X}_{\text{st}})$;
4 ▷ Feature Projection:
5 $\widetilde{F}_{\text{st}}, \widetilde{F}_{\text{text}}, \widetilde{F}_{\text{img}} \leftarrow \text{Project}(F_{\text{st}}, F_{\text{text}}, F_{\text{img}})$;
6 ▷ Cross-modal Attention:
7 $Attn_{\text{st}\rightarrow\text{text}}, Attn_{\text{st}\rightarrow\text{img}} \leftarrow \text{CMA}(\widetilde{F}_{\text{st}}, \widetilde{F}_{\text{text}}, \widetilde{F}_{\text{img}})$;
8 ▷ Fusion:
9 $F_{\text{fused}} \leftarrow \text{Fuse}(\widetilde{F}_{\text{st}}, Attn_{\text{st}\rightarrow\text{text}}, Attn_{\text{st}\rightarrow\text{text}})$;
10 ▷ Graph Construction:
11 $\mathbf{A} \leftarrow \lambda\mathbf{A}^{(0)} + (1 - \lambda)\mathbf{A}^{\text{SHAP}}$;
12 **Function** STED$(X, \mathbf{A})$ :
13     $X_{\text{spatial}} \leftarrow \text{GCN}(X, \mathbf{A})$;
14     $X_{\text{temporal}} \leftarrow \text{Mamba}(X)$;
15     $X_{\text{encoded}} \leftarrow \text{LayerNorm}(X_{\text{spatial}} + X_{\text{temporal}})$;
16     $\hat{Y} \leftarrow \text{MLP}(X_{\text{encoded}})$;
17     **return** $\hat{Y}$;
18 ▷ Dual-branch Training:
19 **for** *each training batch* **do**
20     $\hat{Y}_{\text{st}} \leftarrow \text{STED}(X_{\text{st}}, \mathbf{A})$;
21     $\hat{Y}_{\text{mm}} \leftarrow \text{STED}(\text{BackdoorAdjustment}(F_{\text{fused}}), \mathbf{A})$;
22     $\hat{Y}_{\text{final}} \leftarrow \text{MLP}(\hat{Y}_{\text{st}}; \hat{Y}_{\text{mm}})$;
23     $\mathcal{L}_{\text{pred}}, \mathcal{L}_{\text{st}}, \mathcal{L}_{\text{mm}} \leftarrow \text{ComputeLosses}(Y, \hat{Y}_{\text{final}}, \hat{Y}_{\text{st}}, \hat{Y}_{\text{mm}})$;
24     $\mathcal{L}_{\text{all}} \leftarrow \mathcal{L}_{\text{pred}} + \beta\mathcal{L}_{\text{st}} + (1 - \beta)\mathcal{L}_{\text{mm}}$;
25     Update model parameters by minimizing $\mathcal{L}_{\text{all}}$;

---

# D   Additional Experiment Details

## D.1   Dataset Details

Table 3 summarizes the main characteristics of the datasets used in our study, including their spatial and temporal coverage, and available modalities.

Table 3: Dataset information

| Dataset | #nodes | Spatial Coverage | Time Interval | Time Range | Modality | | |
|---|---|---|---|---|---|---|---|
| | | | | | Time Series | Image | Text |
| Terra | 100 | United Kingdom | 3 hours | 01/1979 - 01/2024 | ✓ | ✓ | ✓ |
| BjTT | 1260 | Beijing | 4 minutes | 01/2022 - 03/2022 | ✓ | ✗ | ✓ |
| GreenEarthNet | 1024 | Global | 1 day | 01/2017 - 12/2020 | ✓ | ✓ | ✗ |
| BikeNYC | 128 | New York | 1 hour | 04/2014 - 09/2014 | ✓ | ✗ | ✗ |

The detailed information of the dataset is as follows:

- **Terra**[4]: Terra is a large-scale, high-resolution, multi-modal dataset that provides hourly spatio-temporal data from 6,480,000 global grid points spanning 45 years, along with supplementary geo-images and text descriptions. In our analysis, we utilize wind speed as the spatio-temporal sequence modality, LLM-generated text descriptions as the text modality, and topographic maps as the image modality. We extract data from the Terra dataset covering the period from January 1979 to January 2024 at 1° spatial resolution, focusing on the United Kingdom with a selection of 100 grid points.

- **BjTT**[59]: BjTT is a public multi-modal dataset for urban traffic prediction, containing 32,400 time-series records of traffic velocity and congestion from 1,260 roads in Beijing's Fifth Ring Road area over three months. The dataset also includes textual descriptions of traffic events such as accidents and roadwork. For our research, we use road velocity as the spatio-temporal modality, event descriptions as the text modality, and extract traffic data at 4-minute intervals from January to March 2022.

- **GreenEarthNet**[3]: GreenEarthNet is a comprehensive, large-scale, multi-modal dataset developed for vegetation estimation using satellite time series data. It comprises spatio-temporal minicubes, each containing a series of 30 satellite images taken at five-day intervals, along with 150 daily meteorological observations and an elevation map. In our work, we adopt the Normalized Difference Vegetation Index (NDVI) as the modality for spatio-temporal sequences, and use satellite images to represent the image modality. Specifically, we crop each spatio-temporal minicube to a resolution of 32 × 32 pixels, corresponding to a 0.64 × 0.64 km area, to reduce computational overhead while preserving essential spatial patterns.

- **BikeNYC**[26]: BikeNYC is a large-scale urban mobility dataset that provides spatio-temporal trajectory data collected from the New York City bike-sharing system in 2014. The dataset includes detailed trip records, comprising trip duration, start and end station identifiers, and corresponding start and end timestamps. For our study, we use bike inflow as the spatio-temporal sequence modality to compare model performance in a single-modal setting.

## D.2   Baselines Description

**(i) Single-modal spatio-temporal prediction methods.**

- **D2STGNN** [40] introduces a novel framework that decouples diffusion and inherent signals in traffic data to improve spatio-temporal prediction.

- **ST-SSL** [35] introduces a spatio-temporal self-supervised learning framework for traffic flow prediction, leveraging adaptive graph-based data augmentation and self-supervised tasks to effectively capture spatial and temporal heterogeneity.

- **HimNet** [10] proposes a meta-learning approach that captures spatio-temporal heterogeneity via learned embeddings to generate adaptive forecasting parameters.

**(ii) Causality-based spatio-temporal prediction methods.**

- **NuwaDynamics** [45] introduces a two-stage causal learning framework that identifies and exploits causal regions in spatio-temporal data to enhance generalization and robustness.
- **CaPaint** [13] proposes a causal framework that leverages diffusion-based inpainting to address non-causal regions and improve generalization in spatio-temporal forecasting.

**(iii) Foundation models for spatio-temporal prediction methods.**

- **GPT-ST** [27] introduces a spatio-temporal masked autoencoder with adaptive masking and hierarchical pattern encoding to pre-train customized representations for improved downstream prediction.
- **UniST** [57] introduces a universal spatio-temporal prediction framework that utilizes pre-training and knowledge-guided prompts to generalize across diverse urban tasks with minimal labeled data.

**(iv) Multi-modal spatio-temporal prediction methods.**

- **T3** [16] proposes a multi-modal traffic forecasting model that fuses pre-trained textual and traffic embeddings to capture event impacts and address data sparsity.
- **From News to Forecast (FNF)** [48] leverages LLM-based agents to integrate and reason over news and time series data, boosting forecasting accuracy through adaptive event incorporation.

## D.3 Implementations

The parameters of all baseline models follow their paper's settings. All experiments are conducted on a Rocky Linux 8.8 server equipped with NVIDIA A40 GPUs. We implement $E^2$-CSTP using Python 3.8.20 and PyTorch 2.0.1. Model training is performed using the Adam optimizer with an initial learning rate of 0.001, which decays by a factor of 0.5 every 5 epochs. We adopt early stopping based on the validation loss with a patience of 10 epochs to prevent overfitting and ensure stable convergence.

## D.4 Metrics

In this appendix, we provide the detailed calculations and interpretations of the evaluation metrics used in this work, including Mean Absolute Error (MAE), Root Mean Square Error (RMSE), and Mean Absolute Percentage Error (MAPE). These metrics are widely adopted in spatio-temporal forecasting tasks for assessing the accuracy of predicted numerical values over time.

MAE measures the average magnitude of the errors between predicted and actual values, without considering their direction. RMSE calculates the square root of the average of squared differences between predictions and actual observations. MAPE expresses the prediction error as a percentage, providing a scale-independent measure of accuracy.

$$\text{MAE} = \frac{1}{N} \sum_{i=1}^{N} |y_i - \hat{y}_i| \tag{29}$$

$$\text{RMSE} = \sqrt{\frac{1}{N} \sum_{i=1}^{N} (y_i - \hat{y}_i)^2} \tag{30}$$

$$\text{MAPE} = \frac{100\%}{N} \sum_{i=1}^{N} \left| \frac{y_i - \hat{y}_i}{y_i} \right|, \tag{31}$$

where $y_i$ is the ground truth, $\hat{y}_i$ is the predicted value, and $N$ is the total number of prediction instances.

## D.5 Model Efficiency Study

Besides the total training time, we also evaluate the per-epoch runtime on 4 datasets under the same batch size (64 for Terra and BikeNYC, 4 for BjTT and GreenEarthNet). Figure 6 further

shows that, on multi-modal datasets, the per-epoch runtime of E$^2$-CSTP is slower than the strongest baseline, which is an outcome that is unsurprising given the extra cross-modal attention layers. By contrast, when trained on single-modal data BikeNYC, the model records a notable 20 % speed-up per epoch compared with the second-best method, confirming that the proposed architectural refinements translate into tangible computational gains in purely single-modal settings.

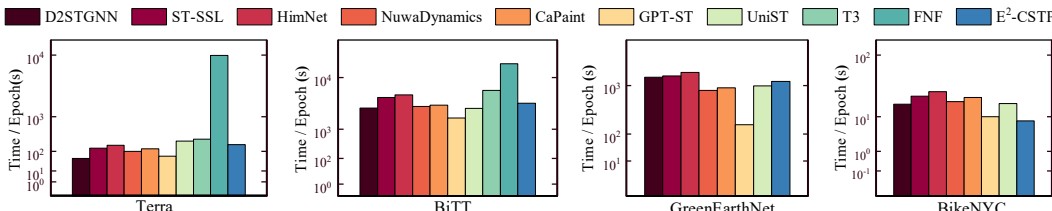

Figure 6: Model efficiency comparison on the per-epoch runtime.

To further evaluate the performance of our prediction module STED, we conduct additional experiments by replacing it with several Transformer-based alternatives—Informer, Autoformer, FEDformer, and iTransformer—on the BjTT, GreenEarthNet, and BikeBYC datasets, in addition to the results presented for Terra in the main text. Each variant is denoted as w/ In, w/ Auto, w/ FED, and w/ iTrans, respectively.

We evaluate both the prediction accuracy (measured by MAE) and computational cost (measured by per-epoch runtime) under the same batch size and training settings for fair comparison. As illustrated in Fig. 7, our proposed prediction module STED consistently achieves better accuracy, with improvements ranging from 15.64% to 44.20%, while also reducing per-epoch runtime by 14.20% to 89.25% across the additional datasets.

These results are consistent with our findings on the Terra dataset and further demonstrate that our prediction module STED offers a favorable trade-off between accuracy and efficiency across diverse spatio-temporal scenarios.

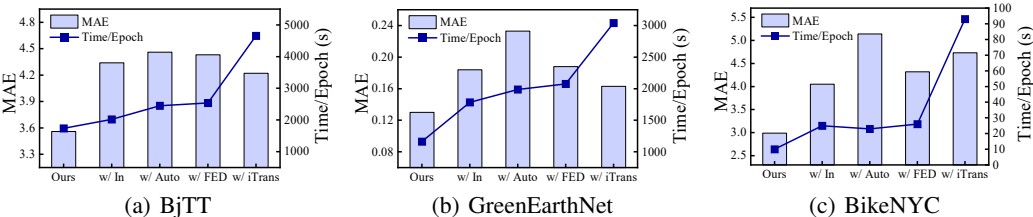

(a) BjTT       (b) GreenEarthNet       (c) BikeNYC

Figure 7: E$^2$-CSTP with Transformer-based variants across datasets.

## D.6 Parameter Sensitivity Study

We evaluate the effects of hyperparameters of the graph fusion factor $\lambda$ among $\{0, 0.25, 0.5, 0.75, 1\}$ and dual-branch loss balancing $\beta$ among $\{0, 0.25, 0.5, 0.75, 1\}$, shown in Fig. 8 and Fig. 9.

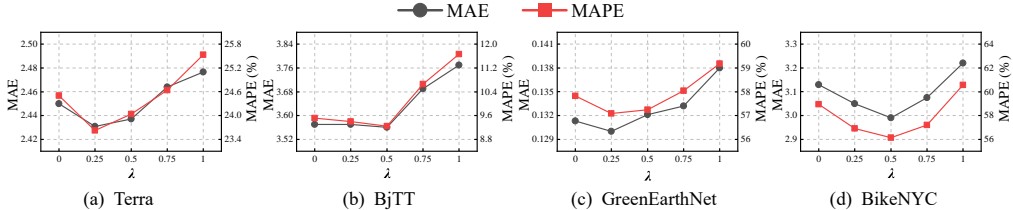

(a) Terra       (b) BjTT       (c) GreenEarthNet       (d) BikeNYC

Figure 8: Parameter sensitivity study on $\lambda$.

The hyperparameter $\lambda$ controls the degree of integration between the spatial adjacency matrix and the causal matrix. A moderate value of $\lambda$ (e.g., 0.25 or 0.5) generally yields better performance, as it balances the influence of raw spatial structure and the refined causal graph.

Specifically, for Terra and GreenEarthNet (remote sensing tasks), a lower $\lambda$ (0.25) works better, which places greater emphasis on the causal graph while reducing the weight of the raw spatial structure. These datasets contain stable spatial correlations (e.g., land use or vegetation types), making the data-driven refinement more valuable than default proximity-based connections. This suggests that in these datasets, dominated by stable and consistent spatial patterns, relying more on causal refinement improves robustness and avoids redundancy. In contrast, BjTT and BikeNYC (urban mobility tasks) benefit from a more balanced integration (0.5), where the influence of the raw spatial and causal graphs is equally weighted. These datasets involve highly dynamic and noisy spatio-temporal flows, requiring a blend of prior structure and adaptive refinement. This indicates that both types of structural information are important for these datasets, likely due to the presence of more dynamic, heterogeneous, or noisy spatial-temporal patterns in urban environments.

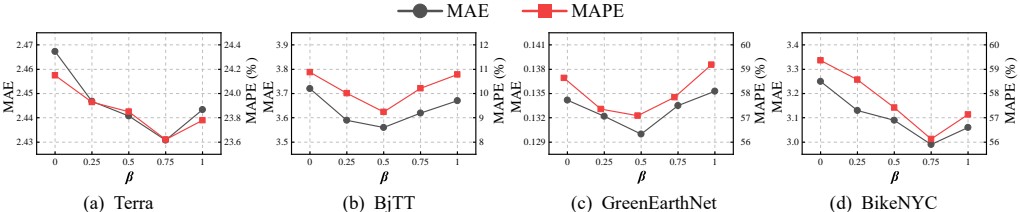

(a) Terra        (b) BjTT        (c) GreenEarthNet        (d) BikeNYC

Figure 9: Parameter sensitivity study on $\beta$.

$\beta$ controls the trade-off between the main and the auxiliary branches, where a larger $\beta$ places more emphasis on the spatio-temporal features, and a smaller $\beta$ increases the relative weight of the auxiliary branch, which incorporates multi-modal data and causal intervention.

For Terra, the optimal $\beta$ is 0.75, highlighting the importance of spatio-temporal modeling in this dataset. GreenEarthNet and BjTT achieve optimal performance with a $\beta$ of 0.5, suggesting a balanced contribution of both branches. In contrast, BikeBYC benefits from a $\beta$ of 0.75. Although this dataset does not include multi-modal inputs, the auxiliary branch still encodes causally refined spatio-temporal features, which contribute to improved prediction performance. Similar reasoning applies to $\beta$, where datasets with stronger exogenous influence (e.g., weather, events) benefit from greater auxiliary-branch emphasis, while those with purer internal spatio-temporal structure favor the primary prediction path.

# E    Limitations and Future Work

E$^2$-CSTP has the potential to benefit critical real-world applications such as intelligent transportation, environmental monitoring, and urban infrastructure management by enabling more accurate and efficient spatio-temporal predictions. While our framework handles spatio-temporal sequence, text and image modalities, other data types such as audio, LiDAR, or social signals (e.g., user interactions) are not considered. Future work could explore a more generalized framework capable of handling a wider range of modalities.

