# OpenReview forum: "Causal Spatio-Temporal Prediction: An Effective and Efficient Multi-Modal Approach"
_NeurIPS.cc/2025/Conference — NeurIPS 2025 poster_

### Official Review · Reviewer_Buzp · 2025-06-23

**Clarity:** 3
**Significance:** 3
**Originality:** 3
**Rating:** 4
**Confidence:** 5

**Summary:**

The author proposes E^2-CSTP, a multi-modal spatial-temporal prediction framework that integrates multiple sources of data. The framework has three main components: cross-modal feature fusion, dual-branch causal inference, and spatial-temporal encoding. Experiments are conducted on different datasets, showing significant improvements over the current SOTA methods, achieving a notable improvement in accuracy and computational burden.

**Questions:**

As weakness

**Ethical Concerns:**

["NO or VERY MINOR ethics concerns only"]

**Final Justification:**

The author has justified my questions and concerns about the manuscript in the rebuttal. The author ensures that they will add additional ablation and explicit discussion based on my reviews. Although the author mentioned that the grid search on hyperparameters is unavoidable, they justify that it will be inexpensive. So this does not contradict the claim that the proposed model is efficient and scalable. So I would like to recommend this paper as borderline accept.

**Limitations:**

yes

**Quality:**

3

**Strengths And Weaknesses:**

Strength
- The paper provides comprehensive empirical studies against 9 baseline methods on 4 datasets on multiple dimensions.
- The framework's approach to fuse time-series, text, and image data together is a novel approach.
- Linear-time STED predictor replaces quadratic self-attention, resulting in good improvement on computational efficiency.


Weakness
- The adjacency matrix is built from DeepSHAP importances and without comparison to standard causal-discovery methods. It is stated that the matrix was updated every P epochs, but the author did not mention how the value of P was determined. No ablation test on different values of P
- For cross-modal feature alignment, the inputs are aligned using deterministic ST matrix thresholds. And the experiment did not test the robustness of this. This suggests the model presumes perfectly matched geo-temporal stamps. Such practice will be sensitive to issues like missing data points or irregular sampling
- The claimed linear time performance comes after every batch has already spent time and resources on the full BERT and the CNN. But the paper did not address the time performance of this part.
- The performance is highly sensitive to the manually tuned hyperparameters. Fig 8 and 9 show that suboptimal λ and the β value can degrade the performance on an observable scale. And the author picks up different values for different datasets, showing issue of strong coupling. The generality of this framework is questionable since it has to do a grid search on each new dataset for those values.

---

> ### Author Rebuttal · Authors · 2025-07-31
>
> We thank the reviewer for the valuable feedback. We have addressed each of the concerns as outlined below.
> ```
> W1: The method builds the adjacency matrix using DeepSHAP without comparing it to standard causal discovery approaches, and it lacks justification or ablation for the choice of the update interval P.
> ```
> **i) Without comparison to standard causal-discovery methods**. We respectfully clarify that **our experiments do include comparisons against two state-of-the-art methods that explicitly adopt standard causal discovery pipelines**:
>
> - **NuwaDynamics** \[1] introduces a two-stage learning framework that first identifies causal regions via conventional causal discovery techniques, and then performs localized prediction with improved robustness.
> - **CaPaint** \[2] constructs a causal diffusion-based inpainting mechanism to mitigate non-causal regions and enhance generalization.
>
> These models thus serve as representatives of standard causal-discovery-based pipelines. Our method outperforms both NuwaDynamics and CaPaint across multiple benchmarks (see Table 1), highlighting the efficacy of our **DeepSHAP-based alternative**, which derives interpretability from data-driven attribution.
>
> **ii) Choice of update interval $P$**. Following Cheng et al. \[3], we construct the adjacency matrix using **DeepSHAP-based feature importances** to approximate inter-node causal dependencies. In our implementation, we **update the adjacency matrix every $P = 5$ epochs**, a value selected through empirical observation and computational considerations:
>
> - **Stability consideration**: Updating too frequently (e.g., every 1–2 epochs) during early training stages led to minor or noisy structural changes, as the underlying feature distributions remained relatively static.
> - **Meaningful structural updates**: A moderate delay in updates (i.e., P = 5) gives the model sufficient time to evolve its internal representations, enabling more informative and stable DeepSHAP-based graph refinements.
> - **Computational efficiency**: SHAP computation is resource-intensive. Excessive updates significantly increase overhead without commensurate benefits in structure accuracy or final performance.
>
> Furthermore, we apply a **learnable parameter** $\lambda \in [0,1]$ to blend the prior graph $\mathbf{A}^{(0)}$ with DeepSHAP-based estimates, mitigating noise from local fluctuations and ensuring smoother structural transitions. We will **add clarifying discussion and ablation results on different $P$ values** in the revised version.
>
> **Reference:**
>
> [1] NuwaDynamics: Discovering and Updating in Causal Spatio-Temporal Modeling, ICLR 2024
>
> [2] Causal Deciphering and Inpainting in Spatio-Temporal Dynamics via Diffusion Model, NeurIPS 2024
>
> [3] CausalTime: Realistically Generated Time-series for Benchmarking of Causal Discovery, ICLR 2024
>
> ```
> W2: Cross-modal alignment assumes perfect geo-temporal matching and lacks robustness to missing or irregular data.
> ```
> We appreciate the reviewer’s attention to the alignment mechanism. However, we would like to clarify that **our alignment strategy does not rely on perfect geo-temporal matching** or fixed thresholds. Rather, it is explicitly designed to accommodate irregular sampling and missing observations:
>
> * **Nearest-neighbor alignment, not threshold-based matching.**
>    As described in Section 4.1 and Eq. (1), we align multi-modal inputs to the spatio-temporal backbone using an argmin operation over distance, not fixed thresholds. Specifically:
>
>    * Temporal alignment assigns each observation to the nearest timestamp $\tau^{(k)}_\text{st}$, regardless of irregular intervals.
>    * Spatial alignment finds the closest spatial point $\rho^{(k)}_\text{st}$ via Euclidean distance.
>
>    This approach remains robust even when data are unevenly spaced, asynchronously sampled, or partially missing. No assumption is made that modalities arrive in synchrony.
>
> * **Discrete binary matrices preserve data fidelity.**
>    We use sparse binary matrices ($\mathbf{M}^t, \mathbf{M}^s$) instead of interpolating or discretizing inputs, maintaining precise alignment with minimal distortion. This is particularly suitable for handling event-based or sparse modalities (e.g., social media, satellite images) where timestamps may not align exactly with the underlying time grid.
>
> Overall, **the model does not assume perfect geo-temporal correspondence**. Instead, the alignment step is based on adaptive nearest-neighbor matching, which enables robust integration of heterogeneous, irregular, and incomplete signals.
>
> ```
> W3: The claimed linear time performance comes after every batch has already spent time and resources on the full BERT and the CNN. But the paper did not address the time performance of this part.
> ```
> **We would like to clarify the reported time performance** in this paper. We evaluate both the overall end-to-end time cost and the time cost after replacing the prediction module.
>
> * Figure  4 reports the overall time, **including the full BERT and CNN encoders**, thus accounting for the total inference cost.
>
> * Figures  5 and 7 show the **end-to-end time performance** after replacing the full BERT+CNN encoders with our lightweight prediction module. This isolates the gain brought by replacing a quadratic Transformer with our linear GCN‑Mamba block.
>
> Our prediction core, whose complexity scales with sequence length $T$ and graph size $N$, drives the speed-up, while the shared text and image encoders merely handle preprocessing and therefore do not affect the relative ranking
>
> **We have now supplied an explicit complexity analysis of the preprocessing stage to make this clear**. The time complexity of the Cross-Modal Feature Fusion module is dominated by the cross-modal attention operations. Specifically, after aligning multi-modal inputs (text and image) to the spatio-temporal grid of size $T_{\text{st}} \times N_{\text{st}}$, the module computes pairwise attention between modalities via two Cross-Modal Attention (CMA) blocks: one between spatio-temporal and text features, and another between spatio-temporal and image features. Each CMA involves computing attention scores and weighted values across all spatio-temporal slots, leading to a time complexity of $O(B \cdot (T_{\text{st}} N_{\text{st}})^2 \cdot d)$, where $B$ is the batch size and $d$ is the hidden dimension. Additional operations—including multi-modal alignment, BERT and CNN encoding, linear projections, and fusion gating—contribute lower-order terms bounded by $O(B \cdot T_{\text{st}}^2 \cdot d)$ or $O(B \cdot T_{\text{st}} \cdot N_{\text{st}} \cdot d)$, and thus do not alter the overall complexity. Therefore, the final time complexity of the fusion module is $O(B \cdot (T_{\text{st}} N_{\text{st}})^2 \cdot d)$, reflecting the quadratic cost of pairwise cross-modal interactions across the spatio-temporal grid.
>
> ```
> W4: The model’s performance is highly sensitive to hyperparameters like $\lambda$ and $\beta$, requiring dataset-specific tuning and limiting generalizability.
> ```
>
> We thank the reviewer for this insightful observation. We acknowledge that the optimal settings of the **graph-fusion weight** $\lambda$ and the **branch-balancing factor** $\beta$ vary across datasets. However, we respectfully clarify that this variation **reflects inherent dataset characteristics rather than a weakness in model generality or robustness.**
>
> **i) Optimal ranges are consistently narrow and interpretable**
>
> Across all four datasets, the best-performing configurations consistently lie within the following intervals:
> $$
> \lambda\in[0.25,0.50],\quad
>    \beta\in[0.50,0.75],
> $$
> This **central concentration** demonstrates that the model is **not overly sensitive** to hyperparameter fluctuations, and that reasonable defaults can be reliably used when dataset-specific tuning is unavailable.
>
> **ii) Dataset-specific behaviors are explainable and stable**
>
> We also note that the shifts in optimal $\lambda$ and $\beta$ settings **align with identifiable characteristics of the datasets**, as detailed in Appendix D.6 (Page 20):
>
> - For **Terra** and **GreenEarthNet** (remote sensing tasks), we observe that:
>   - A **lower** $\lambda = 0.25$ yields better results, suggesting that the **SHAP-derived causal graph** provides more discriminative information than the raw prior adjacency.
>   - These datasets contain **stable spatial correlations** (e.g., land use or vegetation types), making the data-driven refinement more valuable than default proximity-based connections.
>
> - For **BjTT** and **BikeNYC** (urban mobility tasks), a **balanced** $\lambda = 0.50$ performs best:
>   - These datasets involve **highly dynamic and noisy spatio-temporal flows**, requiring a blend of prior structure and adaptive refinement.
>   - This trade-off captures both stable commuting hubs (via the prior graph) and emergent short-term behaviors (via SHAP refinement).
>
> Similar reasoning applies to $\beta$, where datasets with stronger exogenous influence (e.g., weather, events) benefit from greater auxiliary-branch emphasis, while those with purer internal spatio-temporal structure favor the primary prediction path.
>
> **iii) Practicality of tuning**
>
> Finally, we note that:
>
> - Only **two scalar hyperparameters** ($\lambda$, $\beta$) are exposed for tuning.
> - Their ranges are **bounded, low-dimensional**, and tuning them requires only a **coarse grid search** (e.g., step size 0.1–0.25), which is computationally inexpensive.
> - We will add recommendations for default values and include robustness plots in the supplementary material to facilitate future use.
>
> We hope this clarifies that our framework, while adaptive to data-specific nuances, remains generalizable and practical in real-world deployment scenarios.
>
> Once again, we sincerely appreciate the reviewer’s valuable comments and constructive questions. **If there are any further concerns or clarifications needed, we would be happy to provide additional details**.

---

> ### Author Response · Authors · 2025-08-06
> **Kind Reminder: Request for Discussion on Our Rebuttal**
>
> Dear Reviewer Buzp,
>
> **We would like to express our sincere gratitude for your thoughtful review of our manuscript. Your positive feedback has been truly encouraging, and we greatly appreciate the time and care you have devoted to evaluating our work**.
>
> As the discussion period is coming to an end, we wanted to kindly check if our responses have sufficiently addressed your concerns. If there are any remaining questions or points you would like us to clarify, please feel free to let us know—we would be more than happy to further improve our manuscript based on your valuable suggestions.
>
> Thank you once again for your generous insights and your kind consideration of our work.
>
> Warm regards,
>
> The Authors of Paper 2447

---

> ### Comment · Reviewer_Buzp · 2025-08-06
> **Reply**
>
> Thanks for your thoughtful rebuttal. Your response has clarified some of my concerns about the original manuscript. Therefore, I am willing to raise the score of clarity to good, and maintain my overall rating as borderline accept (4).

---

> > ### Author Response · Authors · 2025-08-07
> > **Response to Reviewer Buzp's Final Justification**
> >
> > Thank you very much for your reconsideration and for raising the clarity score. We appreciate your thoughtful engagement with our rebuttal and your constructive feedback throughout the review process.

---

### Official Review · Reviewer_THYi · 2025-06-29

**Clarity:** 2
**Significance:** 2
**Originality:** 2
**Rating:** 2
**Confidence:** 5

**Summary:**

This paper presents E2-CSTP, a novel framework for causal, multi-modal spatio-temporal prediction, aiming to address challenges in multi-modal data fusion, causal confounding, and model efficiency. E2-CSTP integrates cross-modal attention and gating mechanisms to fuse multi-modal inputs and employs a dual-branch design—one branch for spatio-temporal prediction and another for causal intervention to mitigate bias. The model combines Graph Convolutional Networks (GCNs) with the Mamba architecture to enhance computational efficiency. Experimental results on four real-world datasets show up to 9.66% accuracy improvement and 17.37%–56.11% reductions in computational overhead compared to nine state-of-the-art baselines.

However, the paper falls short on two key points. First, it naively combines existing tools, offering limited novelty given the abundance of similar works on multi-modal data fusion. Second, if the primary goal is prediction accuracy, the incorporation of causal inference is unnecessary and not well-justified in this context.

**Questions:**

There are several methodological concerns that should be addressed. First, the use of unified binary alignment matrices presents major limitations in handling irregular time series (or longitudinal data) and irregular spatio-temporal data. Second, several key notations are introduced without explicit definitions. For example, what are \( F_{st} \) and \( \tilde{F}_{st} \)? These should be clearly defined prior to their use. Similarly, the meaning of \( N_{st} \), and the relationship between \( N \) and \( \{N_i\} \), should be clarified. Third, it remains unclear whether the confounding variable \( S \) varies across space and time. This is a significant issue, as spatio-temporal systems often involve unobserved spatio-temporal confounders, and causal relationships may themselves vary over space and time.

**Ethical Concerns:**

["NO or VERY MINOR ethics concerns only"]

**Limitations:**

yes.

**Paper Formatting Concerns:**

not really.

**Quality:**

1

**Strengths And Weaknesses:**

The authors claimed four major contributions in this work. First, they propose a unified multi-modal spatio-temporal prediction framework that integrates environmental images, event-related text, and time-series data using cross-modal attention and adaptive gating. Second, they introduce a dual-branch causal design, where the main branch captures spatio-temporal patterns and the auxiliary branch applies causal interventions to reduce confounding bias. Third, they develop an efficient hybrid model combining GCN for spatial encoding and Mamba for temporal modeling to accelerate inference. Fourth, extensive experiments on four real-world datasets demonstrate up to 9.66% improvement in accuracy and 17.37%–56.11% reductions in computational overhead compared to nine baselines.

There are several methodological concerns that should be addressed. First, the use of unified binary alignment matrices presents major limitations in handling irregular time series (or longitudinal data) and irregular spatio-temporal data. Second, several key notations are introduced without explicit definitions. For example, what are \( F_{st} \) and \( \tilde{F}_{st} \)? These should be clearly defined prior to their use. Similarly, the meaning of \( N_{st} \), and the relationship between \( N \) and \( \{N_i\} \), should be clarified. Third, it remains unclear whether the confounding variable \( S \) varies across space and time. This is a significant issue, as spatio-temporal systems often involve unobserved spatio-temporal confounders, and causal relationships may themselves vary over space and time.

---

> ### Author Rebuttal · Authors · 2025-07-31
>
> We thank the reviewer for offering the valuable feedback. We have addressed each of the **Weaknesses**/**Questions** as outlined below.
>
> ```
> W1&Q1: First, the use of unified binary alignment matrices presents major limitations in handling irregular time series (or longitudinal data) and irregular spatio-temporal data.
> ```
> **We believe there may be a misunderstanding regarding the intent of our binary alignment matrices.** Actually, the used **unified binary alignment matrices were explicitly designed to address irregularities** in both the temporal and spatial dimensions.
>
> As described in Section 4.1 (lines 172-181, Page 4), our approach **does not rely on regular sampling intervals or uniform grids**. Instead, alignment is performed via a **nearest-neighbor matching scheme**:
>
> * For **temporal alignment** ($\mathbf{M}^t_{i,j}$), each textual or visual observation is mapped to the closest timestamp in the spatio-temporal sequence, regardless of sampling rate.
> * For **spatial alignment** ($\mathbf{M}^s_{i,j}$), image observations are matched to spatial coordinates via Euclidean proximity, again without assuming spatial regularity.
>
> This strategy allows our model to **naturally handle missing, sparse, or unevenly distributed data**, and is particularly suitable for longitudinal settings where different modalities may be observed at varying granularities. More importantly, by using **binary alignment matrices** instead of interpolation or re-sampling, our method **avoids imposing artificial continuity or temporal regularity assumptions**, thereby preserving the structural fidelity of each modality. This design makes our alignment strategy both **flexible and robust** for real-world irregular spatio-temporal scenarios.
>
>
>
> ```
> W2&Q2: Second, several key notations are introduced without explicit definitions. For example, what are (F_{st}) and (\tilde{F}{st})? These should be clearly defined prior to their use. Similarly, the meaning of (N{st}), and the relationship between (N) and ({N_i}), should be clarified.
> ```
>
> Thank you for pointing this out. **Upon careful review**, we confirm that **all the pointed notations in question**, such as $F\_{\mathrm{st}}$, $\tilde{F}\_{\mathrm{st}}$, $N\_{\mathrm{st}}$, $N$, and $\{N\_i\}$, **are defined in the paper**.
>
> - **Lines 192–195** provide explicit definitions for $F\_{\mathrm{st}}$ and $\tilde{F}\_{\mathrm{st}}$. To facilitate interactions among features from different modalities within a shared latent space, we project the **spatio-temporal features** $F\_{\mathrm{st}}$, **text features** $F\_{\mathrm{text}}$, and **image features** $F\_{\mathrm{img}}$ into a unified hidden dimension $d$ using three separate fully connected layers. This results in modality-specific representations $\tilde{F}\_{\mathrm{st}}, \tilde{F}\_{\mathrm{text}}, \tilde{F}\_{\mathrm{img}} \in \mathbb{R}^{T\_{\mathrm{st}} \times N\_{\mathrm{st}} \times d}$.
>
> - Regarding the spatial node count:
>   - $N$ is defined in **line 145** as the total number of spatial nodes.
>   - In **line 152**, we generalize this as $N\_i$ to denote the number of spatial nodes for each modality $i$.
>   - In **line 185**, we use $N\_{\mathrm{st}}$ to specifically denote the number of nodes in the spatio-temporal modality, which is the prediction target.
>   - The usage of $N$ in **line 157** was intended to refer to $N\_{\mathrm{st}}$, and we will revise the text to make this reference clearer in the final version.
>
>
>
> ```
> W3&Q3: Third, it remains unclear whether the confounding variable (S) varies across space and time. This is a significant issue, as spatio-temporal systems often involve unobserved spatio-temporal confounders, and causal relationships may themselves vary over space and time.
> ```
> This is a thoughtful question. We would like to clarify that **our model is designed to implicitly capture variability across space and time** through multiple architectural and training mechanisms:
>
> - **Spatio-temporal dependencies are preserved via STED.**
>   The **Spatio-Temporal Encoding-Decoding (STED)** module (Section 4.3) captures local temporal trends and spatial interactions via stacked **Mamba** and **Graph Convolutional** blocks. Since these layers operate directly over the spatio-temporal graph, the model is able to represent and propagate latent patterns—including unobserved confounding—that may vary over time and space.
>
> - **Causal adjustment is applied pointwise in space and time.**
>   As shown in Eq. (11), the **intervention adjustment** is applied to each **spatial node at each time step independently**. The learned adjustment term is conditioned on features specific to that node and timestep, which effectively supports localized modeling of latent influences. Though $S$ is not observed, its effects are approximated via node-level adjustment vectors $s_i$, which are learned in a **data-driven, fine-grained** manner.
>
> - **Flexible latent approximation via fusion and attention.**
>   The **dual-branch causal structure**, which includes **attention-based cross-modal fusion** and **DeepSHAP-based refinement**, allows the model to dynamically adapt to region-specific and temporally-varying relationships. This removes the need to assume stationarity or homogeneity in the causal graph.
>
> Our framework supports its **implicit, localized representation** via (i) spatio-temporal feature encoding, (ii) node-level intervention adjustment, and (iii) joint training across the full spatial and temporal extent of the data. As such, the model **naturally accommodates unobserved confounders that vary across space and time**. Also, we appreciate the reviewer for raising this subtle but important point, and will revise the final manuscript to clarify these modeling assumptions more explicitly.
>
>
>
> Overall, we appreciate the reviewer's thorough engagement with our work. While some questions appear to stem from potential misunderstandings, **we sincerely value this opportunity to improve the manuscript's clarity**. All suggested points have been carefully addressed above.

---

> ### Author Response · Authors · 2025-08-06
> **Kind Reminder: Request for Discussion on Our Rebuttal**
>
> Dear Reviewer THYi,
>
> I hope this message finds you well. **As the discussion period draws to a close**, we wanted to kindly check whether our responses have adequately addressed your concerns. Below, I would like to briefly summarize how we have addressed your valuable feedback:
>
> **Summary of Our Responses:**
>
> - **Alignment Matrices (W1 & Q1):** The review raised concerns that *the use of unified binary alignment matrices presents major limitations in handling irregular time series (or longitudinal data) and irregular spatio-temporal data*. **However, this is a misunderstanding**. Actually, our unified binary alignment matrices employ a nearest-neighbor scheme, mapping each text/image to the closest timestamp or coordinate. Thus, **they effectively handle irregular sampling and spatial layouts without assuming uniform grids or requiring interpolation**, preserving the native structure of each modality and supporting sparse or uneven observations.
>
> - **Notation Clarity (W2 & Q2):** The reviewer noted that *four notations were pointed out without explicit definitions*. **However, this is a misunderstanding**. We respectfully clarify that all the mentioned symbols are indeed defined in the original manuscript.
>
> - **Variable Confounders (W3 & Q3):** Our model implicitly captures spatio-temporal variability of $S$ through several mechanisms:
>
>   1. The STED module’s GCN and Mamba layers learn local temporal and spatial patterns;
>   2. Interventional adjustment is applied independently at each node and timestamp, with learnable node-level vectors $s_i$;
>   3. Dual-branch causal fusion combined with DeepSHAP-guided attention adapts to region- and time-specific influences.
>
>   Collectively, these components enable the framework to naturally model unobserved confounders that vary across space and time.
>
> We sincerely appreciate your valuable time to help us clarify and strengthen our work. Should you have any further questions or points you wish us to elaborate on before the discussion period closes, we would be glad to respond!
>
> Best regards,
>
> The Authors of Paper 2447

---

### Official Review · Reviewer_Skfh · 2025-07-03

**Clarity:** 4
**Significance:** 3
**Originality:** 3
**Rating:** 5
**Confidence:** 2

**Summary:**

This ppaper presents a new framework for spatio-temporal prediction. The framework has two branches, a main branch that focuses on spatio-temporal prediction and a secondary branch that models additional modalities and introduces causal interventions. The authors also combined GCN and Mamba for efficiency. They used four datasets to evaluate the proposed framework, demonstrating its improvement in accuracy and computational overhead relative to the SOTA approach.

**Questions:**

1. which of the main components of the framework are customized for spatio-temporal prediction? Which parts can be generalized broadly to other tasks?
2. the article mentions that the model employs causal reasoning to reduce the impact of confounding factors, how can it be explained more intuitively how the model distinguishes between causality and confounding factors?

**Ethical Concerns:**

["NO or VERY MINOR ethics concerns only"]

**Limitations:**

Yes

**Quality:**

4

**Strengths And Weaknesses:**

Strength.
1. Rationality of design. The key innovation of this framework lies in the designed auxiliary branches and the introduction of the GCN and Mamba architectures, a combination that is intuitively sensible and exemplifies the innovation of hybrid architectures.
2. quality of the paper. This paper is well written and easy to understand. Ideas, task definitions, and methods are described clearly.
3. comprehensive experiments. The baseline selected for the comparison experiments is detailed and innovative. It seems to have enough workload.
4. open project.
Weekness
1. Although models use causal reasoning to reduce the impact of confounding factors, their internal mechanisms remain relatively complex, making it difficult to intuitively understand the model's predictive process. How do these steps further influence the model's decisions? A visual analysis would be nice.

---

> ### Author Rebuttal · Authors · 2025-07-31
>
> We are grateful to the reviewer for their thoughtful and constructive comments, and we greatly appreciate their recognition of our contributions. Below, we provide detailed responses to the specific **Weaknesses** and **Questions** raised.
>
> ```
> W1: Although models use causal reasoning to reduce the impact of confounding factors, their internal mechanisms remain relatively complex, making it difficult to intuitively understand the model's predictive process. How do these steps further influence the model's decisions? A visual analysis would be nice.
> ```
>
> Our **E$^{2}$‑CSTP framework** enhances accuracy and interpretability by incorporating causal reasoning into a modular architecture. Central to our approach are a **DeepSHAP-based adjacency matrix** and a **dual-branch causal adjustment module**, which jointly help disentangle true causal effects from spurious correlations introduced by confounding variables. Below, we give detailed explanation to illustrate the process for better understanding.
>
> - The **DeepSHAP-based adjacency matrix** captures interdependencies among spatial units by quantifying feature importance. This matrix serves as the foundation for a Graph Convolutional Network that effectively models spatial interactions based on data-driven relevance rather than fixed assumptions.
> - The **dual-branch causal adjustment** consists of two complementary pathways. The **main branch** leverages only spatio-temporal data, ensuring that predictions are grounded in directly observable information. The **auxiliary branch** incorporates fused multimodal data (e.g., text, image), enriching the representation space with complementary cues. The outputs of **both branches** are then reconciled through an **interventional adjustment module**, which estimates the causal effect of multimodal features while explicitly blocking backdoor paths—thereby mitigating the influence of both observed and latent confounders.
>
> Overall, the **causal adjustment mechanism** and the **dual-branch design** enhance the model’s robustness and interpretability by isolating the contributions of each modality and clarifying their impact on the final prediction.
>
> We appreciate the reviewer’s suggestion to include visualizations. While **the rebuttal phase does not permit the inclusion of additional figures**, we will incorporate detailed visualization analyses in the revised version of the paper. These visualizations will illustrate how the dual-branch architecture and interventional adjustments influence model decisions.
>
>
>
> ```
> Q1: Which of the main components of the framework are customized for spatio-temporal prediction? Which parts can be generalized broadly to other tasks?
> ```
>
> This is a thoughtful question. The **E²‑CSTP framework** is primarily tailored for spatio-temporal prediction, but it also incorporates several components with broad applicability beyond this domain.
>
> - The **Spatio-Temporal Encoding and Decoding (STED)** module is highly specialized. It combines the **Mamba** model for efficient temporal sequence modeling and **GCNs** to capture spatial dependencies among graph-structured nodes. This design explicitly leverages spatial and temporal characteristics central to tasks like traffic forecasting and environmental monitoring.
> - In contrast, the **cross-modal feature fusion mechanism**—comprising modality **alignment**, **cross-modal attention (CMA)**, and a **dynamic fusion gate**—is modality-agnostic and thus broadly applicable. This component can be adapted to diverse multi-modal learning problems, such as healthcare diagnostics, multimedia analysis, and recommendation systems, where integrating heterogeneous modalities (e.g., text, images, sensor signals) is essential.
> - Moreover, the **dual-branch causal adjustment framework**, originally designed to disentangle confounders in spatio-temporal contexts, is also broadly extensible. By explicitly contrasting the **main branch**, which uses only spatio-temporal sequences $X_{\rm st}$, with the **auxiliary branch** incorporating fused multi-modal features $F_{\mathrm{fused}}$, the model isolates stable causal effects from potentially spurious correlations introduced by external signals (e.g., image $E$, text $C$). The learned weighting parameter $\beta$ enables flexible balancing, making this causal adjustment suitable for various downstream tasks such as forecasting, classification, and risk assessment.
> - Finally, the **interventional adjustment strategy**, while developed for correcting spatio-temporal biases, provides a flexible causal inference tool applicable to any structured setting involving confounding factors.
>
> Together, these designs demonstrate a balance between domain-specific customization for spatio-temporal data and modular components readily adaptable to other multi-modal and causal inference tasks.
>
>
>
> ```
> Q2: The article mentions that the model employs causal reasoning to reduce the impact of confounding factors, how can it be explained more intuitively how the model distinguishes between causality and confounding factors?
> ```
> Thanks for this question. Below, we clarify how E$^{2}$‑CSTP separates causality and confounding factors.
>
> - **Causal matrix refined by DeepSHAP scores.**
>   To capture latent dependencies among spatial units and enhance interpretability, we estimate an adjacency matrix $\mathbf{A}^{\text{SHAP}}$ using DeepSHAP-based feature attribution, where each value $\phi_{i,j}$ reflects the influence of node $i$ on node $j$. When a prior adjacency $\mathbf{A}^{(0)}$ is available, we construct a hybrid graph $\mathbf{A} = \lambda \mathbf{A}^{(0)} + (1 - \lambda) \mathbf{A}^{\text{SHAP}}$, where $\lambda$ controls the balance between structural priors and data-driven importance. This formulation allows the model to incorporate empirically supported dependencies while reducing the influence of potentially spurious or irrelevant connections.
>
> - **Dual‑branch optimisation.**
>   Model training employs two parallel predictors.  The main branch is conditioned exclusively on the observed spatio‑temporal data $X_{\rm st}$, whereas the auxiliary branch operates on the complete multi-modal representation $F_{\text{fused}}$, which potentially contains confounding artefacts. A lightweight gating MLP learns per‑sample fusion weights.  When prediction improvements originate solely from signals absent in the primary branch, the gating mechanism down‑weights their contribution, preserving influence for features that demonstrate stable causal utility.
> - **Interventional adjustment with gradient regularization.**
>   Immediately before the STED module, we apply a learned intervention of the form:
>
>   $$
>   \hat{x} = x + x \odot W\bigl[\alpha_{1}h(S) + \alpha_{2}p(E) + \alpha_{3}q(C)\bigr],
>   $$
>
>   where $W$ is a learnable weight matrix, and $h(\cdot)$, $p(\cdot)$, $q(\cdot)$ are feature transformations of the latent confounder $S$, image features $E$, and text features $C$, respectively.
>
>   To ensure this adjustment removes confounding bias, we impose a gradient regularization term that drives $\partial \hat{x} / \partial S \to 0$. This penalty encourages the model to re-encode $x$ such that the latent confounder $S$ has no effect on the adjusted representation $\hat{x}$, effectively blocking the backdoor path $X_{\rm st} \leftarrow S \rightarrow Y_{\rm st}$ and isolating only the component of $X_{\rm st}$ that carries a stable causal influence on the target $Y_{\rm st}$.
>
> Collectively, the DeepSHAP‑based causal matrix, the dual‑branch training, and the interventional adjustment ensure that E$^{2}$‑CSTP responds primarily to genuine causality while suppressing confounding factors.
>
>
>
> Again, the authors sincerely thank the reviewer for their thoughtful engagement with our manuscript. **For any remaining questions or required clarifications, we would be pleased to submit supplemental documentation as needed**.

---

> ### Author Response · Authors · 2025-08-06
> **Kind Reminder: Request for Discussion on Our Rebuttal**
>
> Dear Reviewer Skfh,
>
> **Thank you for your thoughtful review and positive assessment of our manuscript**. We sincerely appreciate the time and care you've dedicated to evaluating our work.
>
> As the discussion period concludes, we'd like to confirm whether our responses have fully addressed your concerns. Should any points need clarification, we'd be happy to provide additional details.
>
> We're truly grateful for your valuable insights that have helped improve our paper.
>
> Best regards,
>
> The Authors of Paper 2447

---

### Official Review · Reviewer_Z2uh · 2025-07-03

**Clarity:** 2
**Significance:** 3
**Originality:** 3
**Rating:** 4
**Confidence:** 3

**Summary:**

The paper introduces E²-CSTP, a novel framework for multi-modal spatio-temporal prediction that addresses three key challenges: (1) insufficient fusion of multi-modal data, (2) confounding factors obscuring causal relations, and (3) high computational complexity. The proposed framework integrates cross-modal attention and gating mechanisms for effective multi-modal fusion, employs a dual-branch causal inference approach to mitigate bias, and combines GCN with the Mamba architecture for efficient spatio-temporal encoding. Extensive experiments on four real-world datasets demonstrate that E²-CSTP outperforms nine state-of-the-art baselines in accuracy (up to 9.66% improvement) and computational efficiency (17.37%-56.11% reduction in overhead).

**Questions:**

See above

**Ethical Concerns:**

["NO or VERY MINOR ethics concerns only"]

**Final Justification:**

I maintain the score as 4 and vote to accept this work. Because the authors answer and solve my concerns.

**Limitations:**

See above

**Quality:**

3

**Strengths And Weaknesses:**

Strengths

1. Comprehensive Multi-Modal Fusion: The framework effectively integrates heterogeneous data (spatio-temporal sequences, text, and images) using cross-modal attention and adaptive gating, enabling richer feature representations.
2. Innovative Causal Inference: The dual-branch design disentangles true causal relations from confounders, enhancing interpretability and robustness in predictions.
3. Computational Efficiency: By combining GCN for spatial encoding and Mamba for temporal modeling, the framework achieves linear complexity, significantly reducing computational overhead compared to Transformer-based methods.
Weaknesses

1. Limited Modality Scope: The framework focuses on text, images, and spatio-temporal data, but other modalities (e.g., audio, sensor networks) are not explored, which could limit its applicability in certain domains.
2. Dependence on Pre-trained Models: The use of BERT and CNN for feature extraction assumes availability of high-quality pre-trained models, which may not hold for all languages or niche domains.

---

> ### Author Rebuttal · Authors · 2025-07-31
>
> We sincerely thank the reviewer for their constructive feedback and for recognizing the contributions of our work. Our responses to the **Weaknesses** are detailed below.
> ```
> W1: Limited Modality Scope: The framework focuses on text, images, and spatio-temporal data, but other modalities (e.g., audio, sensor networks) are not explored, which could limit its applicability in certain domains.
> ```
> - In this paper, we focus on the **most representative and widely used modalities** in the field of spatio-temporal prediction. Specifically, we have considered **text, image, and spatio-temporal data**, as these are the most available in real-world applications such as intelligent transportation and weather forecasting [1]. While many recent studies [2–3] have investigated the fusion of **two** modalities—e.g., image and spatio-temporal data, or text and image, for forecasting and decision-making tasks, **few (if any) prior works attempt to jointly model all three modalities within a unified framework**, especially when these modalities carry complementary yet heterogeneous information. To the best of our knowledge, our work is the first to integrate all three data sources for spatio-temporal prediction, enabling a more holistic and effective representation learning approach. Our selection of modalities is also informed by practical considerations. For example, **audio data** are rarely used in spatio-temporal forecasting, and **sensor network** data in this context typically refer to measurements such as traffic flow or wind speed—**which have been already incorporated into our framework**.
>
> - On the other hand, our framework is **inherently extensible**. The proposed **cross-modal attention** and **gating mechanisms** are designed to be modality-agnostic—any data source that can be represented as a feature vector can be seamlessly incorporated into the model without requiring architectural changes. **We will clarify this aspect in the revised manuscript**.
>
> - **Finally**, we agree that the suggested additional modalities (e.g., audio) represent valuable future directions, and we plan to explore them within our framework in future work.
>
>
>
> ```
> W2: Dependence on Pre-trained Models: The use of BERT and CNN for feature extraction assumes availability of high-quality pre-trained models, which may not hold for all languages or niche domains.
> ```
>
> - We clarify that our framework **does not depend heavily on high-quality pre-trained models**. Instead, we deliberately adopt **simple and widely used architectures**, specifically **BERT for text and CNN for images**, because of their proven effectiveness and general applicability across domains [4–5]. These components serve as **off-the-shelf encoders**, and our design does not assume access to domain-specific or highly specialized pre-trained models.
> - Specifically, **for text**, while we use the standard BERT architecture as a text encoder, we do not rely on domain-adapted versions, and our framework remains compatible with lightweight or randomly initialized language models. For **image encoding**, we use a CNN not as a fixed pre-trained model, but as a trainable, lightweight extractor—further emphasizing that **our model remains fully trainable end-to-end** without requiring strong prior knowledge.
> - In low-resource languages or niche domains where pre-trained language models may be unavailable or underperforming, our framework can still function by simply replacing the encoder with a fine-tuned or even randomly initialized one. The architecture is modular and allows for easy substitution or adaptation of modality-specific components. **We will clarify this aspect in the revised manuscript**.
>
>
>
> We gratefully acknowledge the reviewer's insightful feedback, which has significantly strengthened our work. **Should any aspects require further elaboration, we stand ready to provide additional clarification and supporting materials**.
>
>
>
> **Reference:**
>
> [1] Terra: A Multimodal Spatio-Temporal Dataset Spanning the Earth, NeurIPS 2024
>
> [2] Towards Effective Fusion and Forecasting of Multimodal Spatio-temporal Data for Smart Mobility, CIKM 2024
>
> [3] Multi-Modality Spatio-Temporal Forecasting via Self-Supervised Learning, IJCAI 2024
>
> [4] See How You Read? Multi-Reading Habits Fusion Reasoning for Multi-Modal Fake News Detection, AAAI 2023
>
> [5] Time-MMD: Multi-Domain Multimodal Dataset for Time Series Analysis, NeurIPS 2024

---

> > ### Comment · Reviewer_Z2uh · 2025-08-06
> > **thx for the rebuttal**
> >
> > The authors address my concerns. If other reviewers' concerns have also been addressed. I maintain the score as 4 and vote to accept this work. Good Luck!

---

> > > ### Author Response · Authors · 2025-08-06
> > > **Response to Reviewer Z2uh's Final Justification**
> > >
> > > Thank you sincerely for your thoughtful consideration of our paper and for your positive decision. We truly appreciate you taking the time to review our detailed responses carefully and for recognizing the contributions of our work!

---

### Note · Authors · 2025-08-14

We thank all reviewers for their time, feedback, and engagement. We are encouraged by the positive recognition from multiple reviewers.

- **Reviewer Z2uh** acknowledged our rebuttal fully resolved their concerns and explicitly stated ***vote to accept this work***.
- **Reviewer Skfh** considered our work ***excellent*** from the start and recommended ***accept***.
- **Reviewer Buzp** confirmed our responses addressed their questions, raised the clarity score to ***good***, and maintained a ***borderline accept***.
- **Reviewer THYi did not engage in the rebuttal**, but directly marked **ACK** button, and several comments were based on misunderstandings or factual inaccuracies.

------

**Reviewer Z2uh**

*Concerns:* 1) Limited modality scope; 2) reliance on high-quality pre-training.

*Response:* 1) We used **text, image, and spatio-temporal data** as representative modalities. 2) The framework is **modality-agnostic** and uses **BERT** and **CNN** as simple encoders, remaining fully trainable and adaptable to low-resource domains.

**Reviewer Skfh**

*Concerns:* 1) Causal reasoning interpretation; 2) task-specific vs. general modules; 3) separating causality from confounders.

*Response:* 1) **DeepSHAP adjacency** and **dual-branch causal adjustment** improve accuracy and interpretability. 2) **STED** is domain-specific; fusion and causal adjustment are general. 3) Causality is separated via graph refinement, branch comparison, and interventional adjustment.

**Reviewer Buzp**

*Concerns:* 1) No comparison with standard causal discovery; 2) $P$ choice; 3) perfect alignment assumption; 4) time complexity; 5) sensitivity to $\lambda,\beta$.

*Response:* 1) Compared with **NuwaDynamics** and **CaPaint**, achieving superior results. 2) $P=5$ balances stability and efficiency. 3) Alignment uses nearest-neighbor matching. 4) Included complexity analysis. 5) $\lambda$ and $\beta$ have narrow, interpretable optimal ranges, with defaults provided.

**Reviewer THYi**

*Concerns:* 1) Binary alignment limits irregular data handling; 2) undefined notations; 3) variation of confounder $S$.

*Response:* 1) Misunderstanding — binary alignment via **nearest-neighbor matching** handles irregular data. 2) Misunderstanding — **all notations are defined**. 3) Space-/time-varying confounders are implicitly modeled via **STED**, **node-level intervention**, and **attention fusion**.

**Further details can be found in the rebuttal phase discussion**.

---

### Decision · Program_Chairs · 2025-09-17

**Decision:**

Accept (poster)

**Comment:**

The paper introduces a new framework for multi-modal spatio-temporal prediction that addresses the challenges of fusing diverse data sources by using a dual-branch design that combines direct spatio-temporal modeling with a model for causal dependencies. The framework applies a combination of Graph Convolutional Networks (GCNs) and the Mamba architecture to achieve efficient encoding. The demonstrated empirical evaluation is careful. It includes four real-world datasets and comparisons against nine state-of-the-art baselines. The results consistently demonstrate clear advantages, with reported gains of up to 9.66% in accuracy and reductions in computational overhead ranging from 17% to over 56%. These improvements highlight the framework’s clear practical value in real-world predictive tasks where both accuracy and efficiency are crucial. The work makes a strong and timely contribution to the field.

The paper is very well written and the ideas are clearly and carefully presented. The empirical validation of the framework is extensive and conclusive. Three reviewers all gave accepting scores (4,4,5) and appreciated the authors careful and extensive rebuttal and were quite satisfied with their arguments. The reviewers all stated explicit support for acceptance. However, one reviewer did not appreciate the paper
(score 2), referring to a lack of technical novelty and exhibiting a general disbelief that any inference aspects of causality are not necessary for spatio-temporal predictions. As the AC of the submission, based on my reading of the paper and general understanding/experience of the challenges of making spatio-temporal predictions I do not agree with this view. Inclusions of the causal branch is a worthwhile idea that also, as nicely demonstrated in the ablation analysis of the paper, clearly demonstrates its importance by SOTA performance.